

# Cloud condensation nuclei phenomenology: predictions based on aerosol chemical and optical properties

Inés Zabala[1,2], Juan Andrés Casquero-Vera[1,2], Elisabeth Andrews[3,4], Andrea Casans[1,2], Gerardo Carrillo-Cardenas[5], Anna Gannet Hallar[5], and Gloria Titos[1,2]

[1]Andalusian Institute for Earth System Research, IISTA-CEAMA, University of Granada, Junta de Andalucía, Granada, 18006, Spain
[2]Department of Applied Physics, University of Granada, Granada 18071, Spain
[3]University of Colorado, CIRES, Boulder, 80309, USA
[4]NOAA, Global Monitoring Laboratory, Boulder, 80305, USA
[5]Department Atmospheric Sciences, University of Utah, Salt Lake City, UT 84112, USA

**Correspondence:** Inés Zabala (ineszabala@ugr.es) and Gloria Titos (gtitos@ugr.es)

**Abstract.** This study presents a comprehensive phenomenological analysis of cloud condensation nuclei (CCN) and aerosol properties — including activation properties, microphysical characteristics, chemical composition, and optical properties — across ten surface sites in different environments. Aerosol properties vary widely, reflecting the diverse environments, and controlling the CCN activation characteristics. Despite their critical role in aerosol–cloud interactions, CCN observations remain

sparse and unevenly distributed, limiting global assessments of activation behavior. To address this gap, this study presents CCN predictive methods based on chemical composition combined with particle number size distribution (PNSD) data, and aerosol optical properties (AOPs). The chemical composition driven predictions are tested using three hygroscopicity schemes. All schemes overpredict the CCN concentrations (median relative bias; MRB=13-15%), although the two composition-derived CCN concentrations are markedly better predictors than the fixed-$\kappa_{chem}$ assumption (MRB=24%). The AOPs-derived CCN

prediction is based on two approaches: an extended empirical parameterization of Shen et al. (2019) (hereafter S2019) to 13 stations, which reduces bias from - 27% to - 8% and improves CCN agreement; and second, a random forest model that infers Twomey activation parameters ($C$ and $k$) using both the S2019 variables and all the available AOPs. Including all AOPs reduces MRB from 19% to 15% and highlights the role of absorption in predicting CCN activation. These findings demonstrate that both chemical and optical measurements can provide a reasonable estimate of CCN concentrations when direct measure-

ments are unavailable. These results enable retrospective analyses of long-term aerosol time series to investigate aerosol–cloud interactions.

## 1 Introduction

Aerosol-cloud interactions (ACI) represent the largest source of uncertainty in quantifying the effective radiative forcing of anthropogenic aerosols, as highlighted in the IPCC (2021) report. Within the total aerosol-induced effective radiative forcing

of $-1.3(\pm0.7)\ Wm^2$, ACI contributes approximately $-1.0(\pm0.7)\ Wm^2$. This substantial uncertainty in ACI related processes



arises primarily from an incomplete understanding of how changes in cloud droplet number concentration and size affect cloud water content and cloud spatial extent. These changes are driven mainly by variations in the abundance of cloud condensation nuclei (CCN) — aerosol particles that act as seeds for cloud droplet activation. Therefore, improving our understanding of CCN variability across spatial and temporal scales is essential to reduce uncertainties in global aerosol–cloud interactions and,
by extension, climate projections (Seinfeld et al., 2016).

Reducing these uncertainties requires an improved understanding of aerosol properties across both long-term/large-scale and short-term/regional contexts. Key properties to reduce these uncertainties include aerosol number concentration, size distribution, chemical composition, and the ability of these particles to act as CCN. Over the past few decades, numerous studies have investigated the spatial and temporal variability of CCN and the factors controlling their concentrations in diverse (urban,
continental, high-altitude, marine, and polar regions) environments (e.g., Ansmann et al., 2023; Deng et al., 2018; Gallo et al., 2023; Jurányi et al., 2011; Patel and Jiang, 2021; Rejano et al., 2021; Rose et al., 2010). However, most of these observations are based on short-term field campaigns and their comparability is limited due to differences in instrumentation and data processing, complicating efforts to quantify CCN impacts at the global scale. Thus, improving our understanding of aerosol-cloud interactions relies heavily on consistent and long-term measurements of particle number size distributions (PNSD), CCN
number concentrations ($N_{CCN}$), aerosol chemical composition and hygroscopicity (Fanourgakis et al., 2019). A significant contribution to addressing this limitation was made by Schmale et al. (2017, 2018), who conducted a phenomenological study of collocated PNSD, chemical composition, and CCN measurements at 11 observatories - eight in Europe, two in Asia, and one in the USA. However, expanding this analysis to a global scale requires a more extensive dataset with measurements in regions not previously studied. To address this, Andrews et al. (2025a) recently compiled a dataset of PNSD, aerosol optical
properties (AOPs), chemical composition and CCN at 10 observatories - three in the continental USA, two in South America, two in the Arctic and two in the middle of the Atlantic Ocean.

Even with the recent improvement in spatial coverage of CCN measurements and harmonized datasets (e.g., Andrews et al., 2025a and others), the limited current availability of direct measurements of $N_{CCN}$ is still not adequate for climate research due to the high spatio-temporal heterogeneity of atmospheric aerosol. To overcome this limitation of regional/short-term mea-
surements, several studies have investigated the use of more widely available aerosol parameters, particularly AOPs, for CCN estimation (e.g., Ghan et al., 2006; Shinozuka et al., 2009; Andreae, 2009; Shinozuka et al., 2015; Jefferson, 2010; Liu and Li, 2014; Tao et al., 2018). These include properties such as the scattering coefficient ($\sigma_{sp}$), back-scattered fraction (BSF), and aerosol optical depth (AOD), which are routinely measured by ground-based networks (e.g., AERONET, GAW) and satellites. For example, Jefferson (2010) used $\sigma_{sp}$, BSF and single scattering albedo (SSA) to parameterize Twomey's empirical CCN
activation parametrization (Twomey, 1959), estimating the coefficients $C$ and $k$. Previous studies have shown that $C$ and $k$ parameterizations are site-dependent and are affected by the loading and chemical composition of aerosol particles, respectively (e.g., Rejano et al., 2021). To address this site dependency, Shen et al. (2019) developed a CCN prediction equation based on in-situ aerosol optical properties and showed that correlations between the fit parameters could be used to reduce site dependency and improve generalization across regions.





The combination of aerosol chemical composition and PNSD within the framework of $\kappa$-Köhler theory has been widely applied
to estimate CCN concentrations (e.g., Cai et al., 2022; Rejano et al., 2024). These estimates rely on different assumptions
regarding the reconstruction of bulk aerosol hygroscopicity from individual chemical components (Schmale et al., 2018; Rejano
et al., 2024). Reported closure agreement varies across studies, with aerosol mixing state identified as a key factor influencing
CCN prediction accuracy (Cubison et al., 2008). The relationship between CCN spectral parameters and aerosol properties is

often highly nonlinear because CCN activation depends not only on particle composition but also on size, with particles of
different diameters activating at different supersaturation (SS) levels (e.g., Liang et al., 2022; Ervens et al., 2007; Nair and Yu,
2020). These nonlinearities limit the effectiveness of traditional linear analyses in fully capturing the complexity of aerosol
CCN activity.

In recent years, machine learning (ML) has emerged as a powerful tool in atmospheric science, capable of capturing complex

nonlinear relationships. To the best of our knowledge, the first application of ML to CCN prediction was introduced by Nair
and Yu (2020) and later expanded by Nair et al. (2020), who developed a model using aerosol chemical composition and me-
teorological parameters under specific SS conditions. Rejano et al. (2024) applied a neural network at a high-altitude site with
four inputs: $N_{80}$ (concentration of particles larger than 80 nm), the OA/PM$_1$ ratio (organic aerosol to PM$_1$ mass concentration),
the oxidation proxy $f_{44}$ (fraction of organic signal at *m/z* 44), and global solar irradiance. Liang et al. (2022) and Lenhardt

et al. (2025) both applied random forest (RF) models, the former achieving robust CCN estimates from AOPs without chemical
data and the latter identifying aerosol size as the main predictor of CCN–lidar backscatter relationships. More recently, Wang
et al. (2025b) applied an ensemble of ML methods to six sites to determine the most important AOPs for CCN prediction.
Collectively, these studies highlight the potential of ML to improve spatial and temporal characterization of CCN, with im-
plications for satellite retrievals and climate models. However, applications remain largely site-specific, and generalizability

across diverse environments is still uncertain, although Wang et al. (2025b) observed consistent patterns within similar site
types.

In this study, observations from 10 observatories comprising collocated measurements of PNSDs, CCN number concentrations,
CCN activation properties, and, in some cases, aerosol chemical composition and AOPs are analyzed. The stations cover a
range of environmental conditions (continental, mountain, marine and polar). In what follows, first, the CCN phenomenology

in terms of CCN concentration and activation parameters related to size distribution information is presented. Next, an overview
of the chemical composition and in-situ AOPs, where available, is presented in connection with the observed CCN properties.
CCN predictions based on aerosol chemical composition are evaluated and two additional approaches using aerosol optical
properties, parameterizations and machine learning, are explored. Finally, the different prediction methods are systematically
compared in the discussion section.



## 2 Methodology

This section first describes the location, environment type and the measurements available for each site. Then a brief description of the data quality control process is given. Next, we describe the CCN activation parameters and AOPs. Several CCN prediction schemes using the chemical composition and AOPs are presented. Finally, the random forest model methodology for CCN prediction is described.

### 2.1 Sites and measurement availability

This study considers 10 sites distributed across various environmental settings. All data presented here are described in Andrews et al. (2025a) and accessible at Andrews et al. (2025b). Figure 1 shows the location, environment and measurement availability of each site, and Tables S2 and S3 in the Supplement present an overview of the characteristics of each station. Three observatories — MAO, COR and SGP — are located in continental environments, with MAO also occasionally influenced by urban emissions from the nearby municipality of Manacapuru (Brazil). Two stations — ASI and ENA — are situated in marine regions (north and south Atlantic Ocean, respectively). Additionally, ANX and MOS are located in the Arctic, where they sample both polar and marine aerosols. The MOS site corresponds to the MOSAiC (Multidisciplinary drifting Observatory for the Study of ArctIc Climate) expedition, where the instruments were deployed on an icebreaker frozen into and moving with the ice (Shupe et al., 2022). The remaining three observatories — GUC, SBS-CP and SBS-SPL — are situated in mountainous terrain in Colorado (USA), although these mountain sites are also subject to continental influences. The SBS-CP and SBS-SPL observations occurred during the STORMVEX (Storm Peak Laboratory Cloud Property Validation Experiment) field campaign (Mace et al., 2010), at the Steamboat Springs Ski Resort, separated by 5 km horizontally and 782 m vertically. The database includes both short-term campaigns with only a few months of measurements and long-term stations with several years of data, such as ENA and SGP. Further details on all sites and campaigns are provided in Andrews et al. (2025a).

From the available dataset developed by Andrews et al. (2025a), the data considered in this study include hourly-averaged measurements of $N_{CCN}$, aerosol activation properties, PNSD, total particle number concentration, chemical composition and AOPs. All data considered have been previously processed, harmonized and quality assured and are freely available (Andrews et al., 2025b). All data are reported at standard pressure and temperature conditions ($T_{std}$=0 °C and $P_{std}$=1013 hPa) and at low relative humidity (<40%) to ensure better comparability of results among collocated instruments at each site and across all 10 stations. The complete processing is described in detail in the data descriptor paper by Andrews et al. (2025a). A brief description of the instruments is provided below.

CCN concentrations were obtained with a CCN counter (CCNC), either the single-column (DMT1C) or the dual-column (DMT2C) version. Both models of CCNC had a column scanning across different SS with time, referred to as column A, and the DMT2C had an additional column measuring at a fixed SS, referred to as column B. Hourly-averaged PNSD data were derived from measurements made with a scanning mobility particle sizer (SMPS). The PNSD files also include the total particle number concentration measured by an independent condensation particle counter (CPC) over the same period. An





integrating nephelometer and a particle soot absorption photometer (PSAP) provided aerosol optical data at most sites. The nephelometer measured aerosol scattering and backscattering coefficients at three wavelengths (450, 550 and 700 nm) and the PSAP measured absorption coefficients at 564, 529, and 648 nm. Optical measurements were made downstream of a switched

impactor system so that both $PM_{10}$ and $PM_1$ values of the optical properties are available. Our analysis primarily relies on hourly $PM_{10}$ optical data, while $PM_1$ absorption data is used to complement the composition data. The chemical composition data sets used in this study consist of hourly measurements from the quadrupole aerosol chemical speciation monitor (Q-ACSM, hereafter referred to as ACSM) and include the sub-micrometer mass concentration of particulate organics, sulfate, ammonium, nitrate, and chloride. Included with the ACSM data is the black carbon mass concentration derived from the $PM_1$

PSAP absorption coefficient at 529 nm.

Tables S2 and S3 provide an overview of the instrument models, available measurements, and site-dependent settings. Note that three (ASI, SBS-CP, and SBS-SPL) and five (ANX, MAO, MOS, SBS-CP, and SBS-SPL) of the 10 sites do not have optical and chemical composition measurements, respectively (Fig. 1).

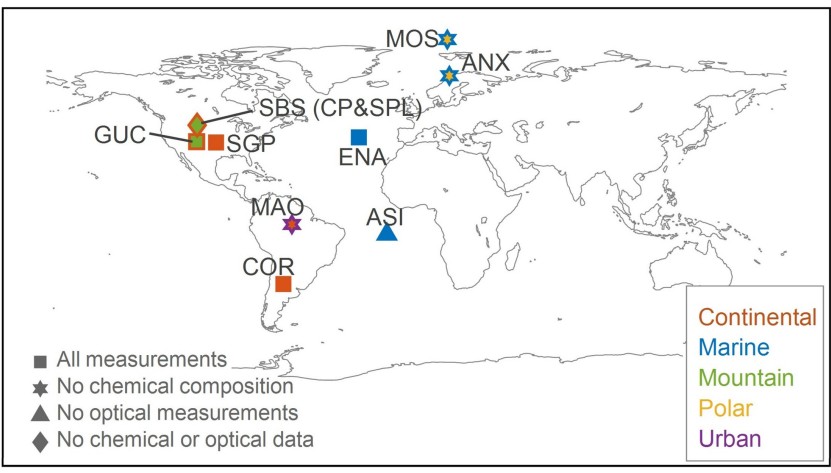

**Figure 1.** Map of sites considered in this study. Site type is indicated with different colors; if the outline is different than the fill color the site could be described by more than one type (e.g., polar and marine). MOS is a mobile deployment so the location represents the midpoint of shiptrack. Symbols indicate measurements availability.

## 2.2 Data quality control

To ensure confidence in the measurements, the datasets used in this study rely on multiple instrument intercomparison quality checks (closure studies) previously described in Andrews et al. (2025a). These checks identify potential inconsistencies between collocated instruments and ensure correct instrument functioning. In this study, we make use of two of these quality checks.



The first quality check applies to DMT2C instruments. CCN concentrations at 0.4% supersaturation measured by column B
are compared with those at the same SS from column A to ensure internal consistency. Data are excluded if the concentration
difference exceeds 50% (quality flag Qc_column_AB in the harmonized files). As shown in Figure S4 of Andrews et al.
(2025a), data from all sites with 2-column CCNC generally show excellent agreement.

The second quality check compares the total particle number concentration ($N_{tot}$) derived from the SMPS PNSD with that
measured by a stand-alone CPC. In this study, SMPS–CPC concentrations are excluded if the relative difference exceeds 50%
(quality check Qc_CPC_SMPS described in Andrews et al. (2025a)), but only when the contribution of particles smaller than
30 nm ($N_{<30}$) to $N_{tot}$ is less than 20% (condition applied in this study). This additional condition avoids removing data due to
discrepancies related to the CPC's lower size cutoff and counting efficiency, especially during new particle formation events,
when CPC counts can substantially exceed those inferred from the SMPS. Overall, the SMPS–CPC comparison across sites
shows good agreement, as illustrated in Figure S1 of Andrews et al. (2025a).

After applying these two quality checks, less than 2% of the CCN column A data and a similarly small fraction of SMPS data
were excluded across all sites. Figure S1 shows the instrument operating periods at each site after these quality checks are
applied. Gaps may also exist due to periods when instruments were offline or not functioning properly, and for optical data,
when sample RH inside the nephelometer exceeded 40%.

For MOS, additional post-processing prior to applying the quality checks was required to remove periods affected by ship
emissions (Boyer et al., 2023), using a pollution detection algorithm previously developed by Beck et al. (2022). The post-
processing pollution detection algorithm was applied to the 5-minute resolution CPC data (MOS_smps_5min in Andrews et al.
(2025b)). As all instruments in this campaign measured from the same inlet, periods identified as polluted using the CPC are
considered polluted for all instruments. The algorithm applies several filters: a power law filter ($a = 0.95$, $m = 0.6$), a threshold
filter ($10$–$10^4$ cm$^{-3}$), a neighboring point filter, a median filter (30, 1.4), and a sparse data filter (30, 24). Only measurements
classified as clean (66% of the original data) are retained. After this filtering, minor additional removal of flagged SMPS
(0.1%) and CCN column A (0.07%) data was applied. Figure S1 shows the available measurement periods at MOS after
applying quality checks and the pollution detection algorithm.

### 2.3 CCN-derived properties

The Andrews et al. (2025a) data sets used in this study also include calculated parameters that can be used to characterize
the CCN activation properties of the aerosol. These parameters are the activated fraction (AF), the critical diameter ($D_{crit}$),
and the hygroscopicity parameter ($\kappa_{CCN}$).The activated fraction (AF) represents the fraction of particles that activate as CCN
at a given SS, calculated as the ratio of CCN concentration to the total particle number concentration. In this study, AF
values derived from CPC measurements were used at all sites except MAO, where SMPS data were used due to the lack
of CPC measurements. The critical diameter ($D_{crit}$) represents the particle size above which all particles are activated into
cloud droplets at a given SS. It can be derived by integrating the PNSD from the largest to the smallest diameters until the
integrated number matches the measured CCN concentration at a given SS (Vogelmann et al., 2012; Jurányi et al., 2011).





Alternatively, if $D_{crit}$ is assumed and size distribution measurements are available but CCN data are not, CCN concentrations can be estimated as the number of particles larger than $D_{crit}$ (Bougiatioti et al., 2009; Kulkarni et al., 2023; Rejano et al., 2024). The hygroscopicity parameter ($\kappa_{CCN}$) quantifies the ability of an aerosol population to absorb water from the environment and

activate as cloud droplets (Petters and Kreidenweis, 2007). $\kappa_{CCN}$ values derived from CCN measurements provide an estimate of the effective hygroscopicity of activated particles in the CCNC and exhibit a dependence on SS. Detailed derivations and equations for these parameters are provided in Andrews et al. (2025a).

### 2.4   ACSM-derived properties

Another approach to estimate the hygroscopicity parameter involves using chemical composition measurements. Since it is

not feasible to determine the properties of each individual particle in the sample, an effective $\kappa_{chem}$ for the entire population is estimated. Petters and Kreidenweis (2007) proposed a simple approximation (Eq. 1) to calculate $\kappa_{chem}$ based on the hygroscopicity parameter ($\kappa$) and the corresponding volume fraction ($\epsilon$) of each species (i) in the sample. This approximation follows the Zdanovskii-Stokes-Robinson (ZSR) approach, assuming a multi-component solution (i.e., a mixture of *n* different solutes) in equilibrium.

$$\kappa_{\text{chem}} = \sum_{i=1}^{n} \epsilon_i \kappa_i, \quad \epsilon_i = \frac{M_i/\rho_i}{\sum_{j=1}^{n} M_j/\rho_j} \tag{1}$$

Here, $M_i$ is the mass of species $i$ and $\rho_i$ its corresponding density. The index $i$ refers to each individual species in the aerosol mixture. The summation in the denominator runs through all species (from 1 to $n$) each time. Further details on the $\kappa_{chem}$ calculation under different assumptions, as well as its use in conjunction with measured size distributions used for CCN prediction, are explained in Sect. 2.6.1.

### 2.5   Optical parameters

The aerosol optical properties can provide insight into the size and chemical composition of aerosol particles. In-situ measurements of multi-wavelength aerosol scattering ($\sigma_{sp}$), back-scattering ($\sigma_{bsp}$), and absorption ($\sigma_{ap}$) coefficients are available at most sites (Tables S2 and S3). From these measurements, several optical parameters were calculated, including the back-scattered fraction (BSF), scattering Ångström exponent (SAE), absorption Ångström exponent (AAE), and single scattering

albedo (SSA) following standard formulations (see Sherman et al., 2015; Shen et al., 2019).

The BSF indicates the relative abundance of smaller particles (D<0.3 $\mu$m) (Collaud Coen et al., 2007), while the SAE describes the wavelength dependence of $\sigma_{sp}$ and serves as an additional proxy for particle size (Seinfeld and Pandis, 1998). BSF and SAE are sensitive to different segments of the aerosol size distribution (Collaud Coen et al., 2007); BSF is more responsive to particles in the lower part of the accumulation mode, whereas SAE is more influenced by particles in the upper part of

the accumulation mode and the coarse mode. The AAE is calculated analogously to SAE and provides insight into aerosol composition, with values near 1 indicating the influence of dust or organic carbon (e.g., from biomass burning) (Bergstrom





et al., 2007; Kirchstetter et al., 2004). The SSA quantifies the relative contribution of $\sigma_{sp}$ and $\sigma_{ap}$ and is also related to particle composition. All optical parameters were calculated at the native instrument wavelengths, except SSA where the absorption was adjusted to 550 nm to match the scattering wavelength: BSF at 550 nm, SAE using 450 and 700 nm wavelengths, AAE with 464 and 648 nm wavelengths, and SSA at 550 nm.

## 2.6 CCN prediction methods

Although CCN concentration measurements are crucial for accurate representation of the CCN availability and variability across sites, these observations are not always available. As noted in the introduction, various methods have been developed to overcome this observational limitation and predict CCN concentrations (e.g. Gysel et al., 2007; Jefferson, 2010; Shen et al., 2019). In this section, we describe the three methods we apply to predict CCN concentration.

### 2.6.1 CCN prediction using chemical composition

CCN concentrations can be predicted using $\kappa$-Köhler theory together with PNSD measurements (Eqs. 3 and 4 in Andrews et al. (2025a)), once the bulk hygroscopicity parameter ($\kappa_{\mathrm{chem}}$) has been derived. Below we describe the three schemes used to calculate $\kappa_{chem}$:

*Scheme 1*: Chemical composition measurements from the ACSM and the BC mass concentration are considered, so Eq. (1) can be expressed in terms of three main components: organics (OA), inorganics (IA), and black carbon (BC) (Eq. 2). This approximation has been shown to provide a reliable estimate of the effective aerosol hygroscopicity (e.g., Bougiatioti et al., 2009; Rejano et al., 2024).

$$\kappa_{\mathrm{chem}} = \kappa_{\mathrm{OA}}\epsilon_{OA} + \sum_{i}(\kappa_{\mathrm{IA}_i}\epsilon_{IA_i}) + \kappa_{\mathrm{BC}}\epsilon_{BC} \tag{2}$$

The contribution of inorganic aerosols to $\kappa_{chem}$ includes several inorganic salts present in the atmosphere, such as ammonium nitrate, ammonium sulfate, ammonium bisulfate and sulfuric acid. The volume fractions of these salts are determined using the simplified ion pairing scheme from Gysel et al. (2007). The densities and $\kappa$ values used for each component are summarized in Table S4 in the Supplement.

*Scheme 2*: To better understand the influence of black carbon on aerosol hygroscopicity, Scheme 2 excludes BC from the $\kappa_{\mathrm{chem}}$ calculation, focusing only on the hygroscopic components (inorganic salts, acids, and organics), which aligns with approaches commonly used in previous literature (e.g., Almeida et al., 2014; Schmale et al., 2018; Rejano et al., 2024). Comparison of both schemes allows for a clearer evaluation of the extent to which BC modulates the overall hygroscopic behavior of the aerosol population.

*Scheme 3*: To complement these two approaches, Scheme 3 is introduced, in which a constant value of $\kappa_{\mathrm{chem}} = 0.3$ is assumed. This scheme aims to serve as a simplified reference, independent of aerosol chemical composition. The value of 0.3 is com-





monly used in the literature as representative of average aerosol hygroscopicity under diverse atmospheric conditions (e.g., Schmale et al., 2018; Pringle et al., 2010). Pringle et al. (2010) report global mean $\kappa_{chem}$ values of 0.27 for continental regions at the Earth's surface, supporting the use of 0.3 as a reasonable approximation for bulk aerosol hygroscopicity.

### 2.6.2 CCN prediction using optical properties

The prediction of CCN concentrations from aerosol optical properties has been explored in several studies (e.g., Ghan et al., 2006; Jefferson, 2010; Shinozuka et al., 2009, 2015; Liu and Li, 2014; Rejano et al., 2021). In addition to exploring the ability of AOPs to estimate CCN concentrations, the main application of this approach is for improving satellite retrievals (e.g., Shinozuka et al., 2015). In Shen et al. (2019) (hereafter referred to as S2019), a new empirical parameterization was developed by analyzing in situ measurements at six stations representing different environments. S2019 investigated the relationships

between CCN concentrations at different SS and AOPs, and derived the following parameterization that explicitly depends on the SAE, BSF, $BSF_{min}$ (1st percentile of BSF data) and $\sigma_{sp}$ of $PM_{10}$ particles:

$$N_{\text{CCN,S2019}}(SS) \approx \left[ (286 \pm 46)\,\text{SAE} \cdot \ln\left(\frac{\text{SS}}{0.093 \pm 0.006}\right)(\text{BSF} - \text{BSF}_{\min}) + (5.2 \pm 3.3) \right] \cdot \sigma_{\text{sp}}. \tag{3}$$

This parameterization is designed to be applicable to any site, regardless of its environmental conditions, and for any SS < 1.1% and provides a basis for estimating $N_{CCN}$ directly from optical measurements (Shen et al., 2019).

In this study, we first test the generality of Equation 3 and assess whether its performance holds across a wider range of aerosol types. Then we apply the S2019 methodology to our 7 sites plus the 6 sites utilized by S2019 to develop a new equation based on 13 sites to see if it improves the predictions of $N_{CCN}$. The derivation is detailed in the Appendix and leads to the following equation:

$$N_{\text{CCN,new}}(SS) \approx \left[ (320 \pm 78)\,\text{SAE} \cdot \ln\left(\frac{\text{SS}}{0.089 \pm 0.011}\right)(\text{BSF} - \text{BSF}_{\min}) + (8.7 \pm 9.3) \right] \cdot \sigma_{\text{sp}}. \tag{4}$$

For the seven sites with available AOPs included in this study, the $BSF_{\min}$ is estimated as $0.11 \pm 0.01$. Accounting for the uncertainties in the regression coefficients, the propagated relative uncertainties in the predicted CCN concentrations are 81%, 34%, 27%, 26%, 25% and 25% at supersaturations 0.1, 0.2, 0.4, 0.6, 0.8 and 1.0%, respectively. Applying the original S2019 parameterization (Eq. 3) to the same dataset yields uncertainties from 16% to 52%. The wider error range in the new fit is driven primarily by the larger standard deviation of $R_{min}$, defined as the first percentile of $N_{CCN}/\sigma_{sp}$ (see Appendix for details),

which is $\pm 9.3 \, \text{cm}^{-3}$ Mm compared to $\pm 3.3 \, \text{cm}^{-3}$ Mm in S2019. It is important to highlight several methodological differences between our approach and that of Shen et al. (2019). Although both studies include measurements from the MAO site, in our analysis this site is treated as independent from that in S2019 due to differences in time periods and data constraints: we used data from 2014–2015 and applied a relative humidity (RH) filter (RH < 40%), while S2019 only used 2014 data without RH restrictions. Similarly, for the ASI site, S2019 included optical measurements acquired at ambient RH > 40%, whereas we



limited our analysis to dry conditions (RH < 40%) and thus did not include ASI data. Furthermore, instead of applying a threshold of $\sigma_{sp}$>10 $Mm^{-1}$ as in S2019, our study used a less restrictive filtering approach by excluding only data ($\sigma_{sp}$, BSF and SAE) when $\sigma_{sp}$ values were below 0.5 $Mm^{-1}$ and above the 99.5th percentile, allowing a broader range of scattering conditions to be considered. Differences in the treatment of CCN data may also contribute to the variability between the resulting parameterizations.

**2.6.3   CCN prediction based on AOPs using the Twomey equation and a random forest model**

The Twomey equation (Twomey, 1959) describes the relationship between supersaturation (SS) and CCN concentration ($N_{CCN}$) via a power law with parameters $C$ and $k$:

$$N_{CCN}(SS) = C \cdot SS^k. \tag{5}$$

This relationship is depicted graphically in Fig. S2 (solid lines) for some of the sites considered here. While Figure S2 shows 265 the overall fits to the data for each site, $C$ and $k$ can also be found for each individual SS scan at each site. Previous studies have found strong correlations between $C$, $k$ and various aerosol properties (Jefferson, 2010; Rejano et al., 2021). Here, machine-learning is applied to predict these parameters from AOPs.

Random forest (RF) is a machine learning method that relates target variables (here, $C$ and $k$) to predictors or "features" (Breiman, 2001; Cutler et al., 2012; Grange et al., 2018). Its main tuning parameters are (a) the number of trees, (b) the number 270 of features considered at each decision node, and (c) the minimum number of observations required in a terminal or "leaf" node (also known as minimum leaf size), which controls the depth and complexity of each tree. The RF model might give better predictions with more trees and more explanatory variables considered, but that also increases the computational cost. Here, we use combinations of AOP variables ($\sigma_{sp}$, $\sigma_{ap}$, BSF, SAE, SSA, and AAE) as predictors to train the model. The RF algorithm is trained on one portion of the data and then the results of the training are applied to the non-training or test data 275 to validate the prediction. In this work, two different validation strategies are considered. First, our primary validation uses a stratified 70 / 30 split: for each site, 70% of scans are randomly chosen for training and the remaining 30% for testing. These per-site subsets are then pooled across all sites to form single training and test sets. Second, as an additional check, we perform leave-one-site-out (LOSO) cross-validation—iteratively holding out one site for testing and training on the others—to assess how including or excluding any given station affects model performance and to verify that the 70 / 30 approach yields valid 280 results across all locations. The predictors are not scaled or normalized before processing.

We implemented RF in MATLAB with TreeBagger function considering 500 trees, using the default minimum leaf size value (1) and sampling all predictors at each split. Performance was assessed via out-of-bag (OOB) error, and feature importance via OOB-permutation (Breiman, 2001). The model was run once to find the features relevant for $C$ and then again, on the same data, to find the features relevant for $k$. Normalized importance scores reveal the variables that most consistently predict $C$ and





$k$. These predicted $C$ and $k$ values are then plugged into the Twomey power-law (Eq. 5) to estimate CCN concentrations at any given SS.

## 3 Results

In this section, we present the results showing the phenomenology of aerosol and CCN activation properties for all the stations considered in this study and the CCN prediction outcomes. We first provide a general overview of aerosol microphysical

and CCN activation properties to demonstrate the range and variability of these characteristics at the 10 sites. Next, we summarize the aerosol chemical composition and use them to predict $N_{CCN}$ for the sites where ACSM data are available using $\kappa_{chem}$. Similarly, we summarize the observed AOPs, where available, and use them to predict $N_{CCN}$, using the S2019 and RF methods. Finally, we evaluate the various CCN prediction methods we have applied and make recommendations for future studies.

### 3.1 Overview of aerosol and CCN activation properties at 10 sites

A summary of aerosol and CCN parameters at 0.4% supersaturation for each site is presented in Figure 2 as normalized frequency distributions. To facilitate a direct comparison with the results of Schmale et al. (2018), the distributions were computed using the same or comparable binning methods and normalized to the total number of data points at each station. However, we focus our analysis on 0.4% SS - rather than 0.2% SS used by Schmale et al. (2018) - because the measurements

at 0.4% SS undergo an additional quality check (see Sect. 2.2), ensuring greater reliability of the data. The leftmost column (Fig. 2a) shows $N_{CCN}$ (colored solid line) overlaid with total particle number concentration ($N_{tot}$, black dashed line). The center column (Fig. 2b) shows $D_{crit}$ (colored solid line) overlaid with the geometric diameter ($D_{geo}$, black dashed line) of the PNSD. The rightmost column (Fig. 2c) depicts the CCN hygroscopicity parameter ($\kappa_{CCN}$). Table 1 provides the median values together with the 25th and 75th percentiles (P25–P75) for the five parameters shown in Fig. 2 and for the activated fraction.

All variables referred to 0.4% SS.

Stations located in polar environments (MOS and ANX) tend to have the lowest $N_{tot}$ and $N_{CCN}$ (Fig. 2a), which is characteristic of the Arctic maritime environment (Barrie, 1986; Schmale et al., 2018). These sites are representative of pristine environments with minimal local sources of aerosols, dominated by natural processes and occasional long-range transport from distant regions. A similar trend was observed in other Arctic sites such as Barrow (Alaska) by Schmale et al. (2018). Slightly

higher $N_{tot}$ and $N_{CCN}$ are observed at the ENA and ASI marine sites compared to the Arctic sites, consistent with these two sites being remote marine locations where aerosols are primarily influenced by natural sources such as sea salt and biogenic emissions (Quinn et al., 2023; Wilson et al., 2015). ENA shows higher concentration of particles, likely associated with local sources due to the proximity of the station to an airport (Gallo et al., 2020). However, CCN concentrations are lower at ENA than at ASI, leading to a smaller activated fraction at ENA (0.26) compared to ASI (0.85). This indicates a lower activation

ability of aerosol particles at ENA. In contrast, the high activated fraction observed at ASI are consistent with Zuidema et al. (2016), who reported that nearly all aerosol particles at this site could activate as CCN even at low SS.



The three mountain sites (GUC, SBS-CP, SBS-SPL) exhibit higher $N_{tot}$ and $N_{CCN}$ at 0.4% SS than the polar and marine sites. SBS-SPL shows the lowest $N_{tot}$ and $N_{CCN}$ of the three mountain sites. SBS-CP is a site where the difference between $N_{tot}$ and $N_{CCN}$ is particularly pronounced, with $N_{tot}$ up to six times larger than $N_{CCN}$. Both distributions are relatively narrow,

suggesting that limited aerosol sources influence the site. The region where SBS-CP is located experiences springtime dust transport from both local and remote sources, which affects overall hygroscopicity (Hallar et al., 2015). Although the SBS-SPL site is very close to the SBS-CP site (SBS-SPL is 5 km east of SBS-CP), the altitude difference (∼2500 m for SBS-CP and ∼3200 m for SBS-SPL) makes SBS-CP more susceptible to influence from the atmospheric boundary layer, while SBS-SPL is more likely to measure free troposphere aerosol in the cooler months when these measurements were made. SBS-SPL is

frequently in-cloud which may also lower aerosol loading via wet scavenging (Hallar et al., 2025). The $N_{CCN}$ distribution at GUC is broader and shows higher concentrations than SBS-SPL despite their similar altitude. This is related to the influence of biomass burning intrusions during June and September 2022 (Gibson et al., 2025) affecting GUC. The three mountain sites show low activated fractions at 0.4% SS (0.11, 0.24 and 0.19, at SBS-CP, GUC and SBS-SPL, respectively) compared to other high-mountain sites (Schmale et al., 2018; Rejano et al., 2021; Jurányi et al., 2011).

Frequency distributions of $N_{tot}$ and $N_{CCN}$ for the continental sites are shifted to higher particle and CCN concentrations. These sites represent regions with a mix of natural and anthropogenic influences, where long-range transport of pollution and local emissions contribute to the aerosol burden. The highest concentration of particles is observed at COR (median value of 3017 cm⁻³, with concentrations above 10000 cm⁻³), which is frequently affected by biomass burning from the Amazon and anthropogenic emissions from Chile and Argentina (Fast et al., 2024). MAO exhibits a broad $N_{CCN}$ and $N_{tot}$ frequency

distribution with an extended tail at the upper end of the distribution. The high $N_{CCN}$ (and $N_{tot}$) values at MAO are associated with the station being affected by the regional transport of biomass burning pollutants (especially in the dry season, July–December) and to the Manaus (city located located 70 km upwind) urban plume (Rizzo et al., 2013). COR and MAO show similar activated fraction of 0.29 and 0.25, respectively. Slightly higher AF is observed at SGP (0.38) associated with higher CCN concentrations.

The center column of Fig. 2 allows us to compare $D_{crit}$ and the size distribution $D_{geo}$ at different sites. $D_{geo}$ serves as a proxy for the aerosol size distribution. Notable differences are observed in both the position and amplitude of the frequency distributions, suggesting variations in aerosol composition and activation processes across locations. Overall, $D_{crit}$ is generally shifted to higher values compared to $D_{geo}$, indicating that a substantial fraction of particles do not reach the CCN activation threshold at 0.4% SS. A similar trend between $D_{crit}$ and $D_{geo}$ was observed at most of the sites analyzed in Schmale et al.

(2018). However, at ASI and MOS, $D_{crit}$ is lower than $D_{geo}$, meaning that at 0.4% SS, most particles activate as CCN. This difference is particularly pronounced at ASI, which is consistent with its high $\kappa_{CCN}$ values (median of 0.75; Fig. 2c, Table 1) and activated fraction (0.85), indicating a predominance of highly soluble aerosols, such as sea salt. Dedrick et al. (2024) showed high hygroscopicity values during clean conditions ($\kappa_{chem} > 0.7$) and lower values during smoke dominated periods ($\kappa_{chem} \sim 0.3$-0.4). Despite also being a marine station, ENA exhibits broader frequency distributions centered on larger values,

with overlapping $D_{crit}$ and $D_{geo}$, suggesting that only a fraction of the particles activate at 0.4% SS (AF median value of 0.26).





This aligns with the wide range of hygroscopicity values observed at ENA, reflecting a mixture of marine aerosols and other sources, likely local emissions such as the nearby airport.

Of the two polar stations, ANX exhibits a lower median $D_{crit}$ (55 nm), indicative of relatively hygroscopic aerosols, whereas MOS shows a higher median value (85 nm). The $D_{crit}$ at MOS is broadly consistent with previous short-term, episodic obser-
vations (Dada et al., 2022), which report $\approx 80$ nm at SS = 0.29% and $\approx 50$ nm at SS = 0.78% under background conditions. At mountain stations, SBS-SPL stands out with the lowest $D_{crit}$ (59 nm) and the highest value of $\kappa_{CCN}$ (0.35), indicating a significant fraction of hygroscopic aerosols. This high hygroscopicity value could be attributed to the influence of anthropogenic $SO_2$ plumes from nearby coal-fired power plants, which have been shown to enhance particle growth from NPF to CCN-relevant sizes and thus facilitate CCN activation at SPL (Hirshorn et al., 2022).

In contrast, SBS-CP exhibits broader distributions and higher $D_{crit}$ values, suggesting a more diverse aerosol mixture influences this site than SBS-SPL. The GUC mountain site exhibits frequency distributions similar to those of continental stations, characterized by $D_{crit}$ distributions shifted toward intermediate-to-high values. The bimodal distribution of $D_{geo}$ observed at GUC suggests the coexistence of different aerosol types, potentially with distinct hygroscopic properties. The first mode, with values lower than $D_{crit}$, likely corresponds to highly soluble particles such as sulfates. In contrast, the second mode, at larger
diameters, may be associated with less hygroscopic aerosols, such as organic compounds related to biomass burning. Among continental stations, SGP has the lowest median $D_{crit}$ (76 nm), indicating a higher fraction of CCN-active aerosols compared to COR (82 nm) and MAO (98 nm). This is consistent with the higher $\kappa_{CCN}$ and activated fraction observed at SGP.





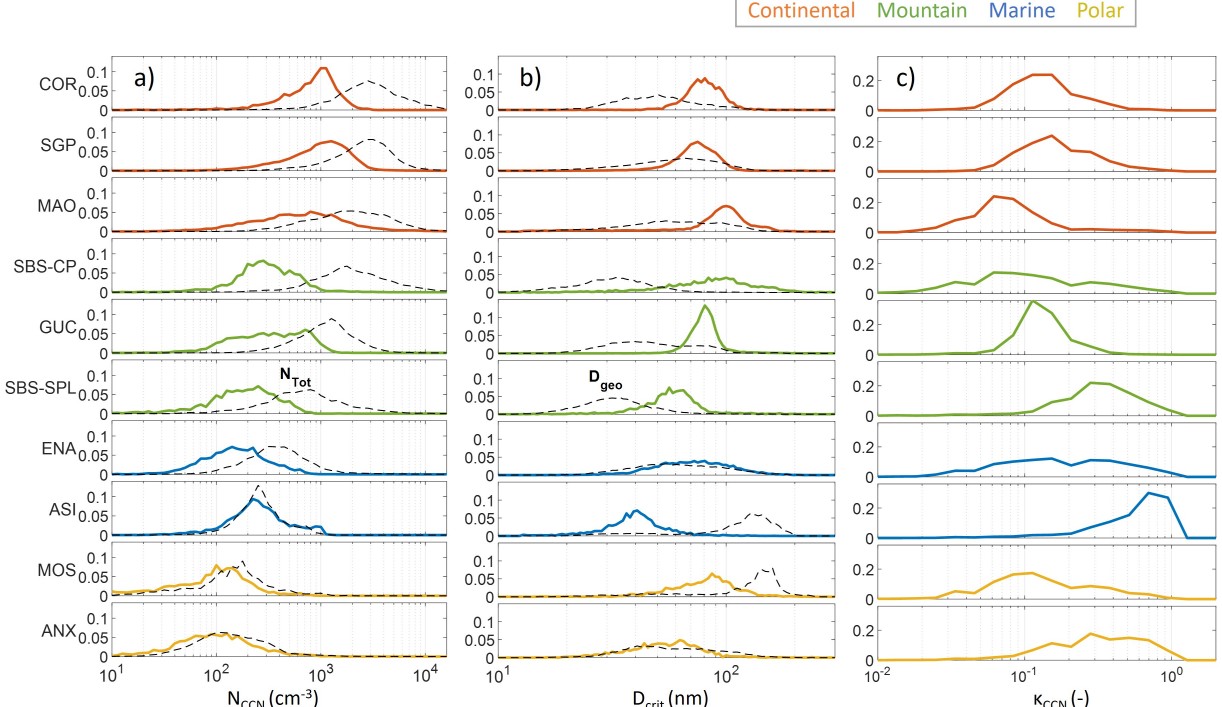

**Figure 2.** Normalized frequency distributions of (a) CCN number concentration ($N_{CCN}$) and total particle concentration ($N_{tot}$) in black, (b) critical diameter ($D_{crit}$) and geometric diameter ($D_{geo}$) in black and (c) hygroscopicity parameter ($\kappa_{CCN}$). All parameters related to CCN measurements are at 0.4% SS.

**Table 1.** Median values and percentiles 25th and 75th (P25–P75) of the total aerosol concentration ($N_{Tot}$), CCN concentration ($N_{CCN}$), geometric diameter ($D_{geo}$), critical diameter ($D_{crit}$), hygroscopicity parameter ($\kappa_{CCN}$) and activated fraction (AF) for each measurement location grouped by site type. All parameters related to CCN measurements are at 0.4% SS.

| Site location | $N_{tot}$ (cm$^{-3}$) | $N_{CCN}$ (cm$^{-3}$) | $D_{geo}$ (nm) | $D_{crit}$ (nm) | $\kappa_{CCN}$ (-) | AF (-) |
|---|---|---|---|---|---|---|
| Continental | | | | | | |
| COR | 3017 (1940-4660) | 927 (589-1222) | 49 (38-64) | 82 (74-91) | 0.15 (0.11-0.20) | 0.29 (0.17-0.43) |
| SGP | 2806 (1790-4035) | 1061 (637-1564) | 61 (44-82) | 76 (66-85) | 0.18 (0.13-0.28) | 0.38 (0.23-0.54) |
| MAO | 2030 (1106-3636) | 659 (325-1253) | 59 (43-85) | 98 (82-113) | 0.08 (0.06-0.12) | 0.25 (0.15-0.42) |
| Mountain | | | | | | |
| SBS-CP | 2011 (1246-3500) | 310 (213-485) | 32 (25-41) | 88 (64-113) | 0.12 (0.06-0.25) | 0.11 (0.05-0.21) |
| GUC | 1195 (780-1698) | 348 (184-637) | 46 (35-66) | 82 (76-88) | 0.15 (0.12-0.18) | 0.24 (0.13-0.40) |
| SBS-SPL | 712 (421-1198) | 193 (115-306) | 33 (27-41) | 59 (51-68) | 0.35 (0.25-0.54) | 0.19 (0.10-0.35) |
| Marine | | | | | | |
| ENA | 398 (259-609) | 160 (101-249) | 61 (44-85) | 74 (55-95) | 0.20 (0.09-0.39) | 0.26 (0.17-0.35) |
| ASI | 271 (205-363) | 255 (178-375) | 126 (95-146) | 41 (35-48) | 0.75 (0.48-1.04) | 0.85 (0.73-0.94) |
| Polar | | | | | | |
| MOS | 156 (94-230) | 103 (48-158) | 140 (98-157) | 85 (66-98) | 0.13 (0.08-0.25) | 0.78 (0.61-0.87) |
| ANX | 138 (86-238) | 100 (58-172) | 57 (41-82) | 55 (43-68) | 0.35 (0.23-0.60) | 0.36 (0.18-0.60) |



## 3.2 Aerosol chemical composition and CCN prediction

### 3.2.1 Overview of aerosol composition

The aerosol sub-micrometer chemical composition measured with the ACSM is available at five of the ten stations (see Tables S2 and S3 for details). The operating temperature of the ACSM (600°C) is not high enough to vaporize refractory components of the aerosol particles, thus only the non-refractory components can be analyzed. As a result, components such as elemental carbon, crustal material, and sea salt cannot be detected (Wu et al., 2016). To complement the ACSM chemistry, BC concentrations are derived from the PSAP absorption coefficient measurements at all sites except ASI, where BC data are not available in

the Andrews et al. (2025b) dataset. Figure 3 presents pie charts that illustrate the relative contribution of the species considered (organics, $SO_4^{2-}$, $NO_3^-$, $NH_4^+$, $Cl^-$, BC) to $PM_1$ at each site, along with the total mean mass concentration.

The mean concentration of $PM_1$ in the five sites ranges from 0.54 to 5.56 $\mu g/m^3$, with varying contributions of the different components, reflecting the distinct aerosol characteristics of each location during the measurement period. Continental sites, COR and SGP, exhibit the highest concentrations (4.01 and 5.56 $\mu g/m^3$, respectively). The mean value measured at SGP is

slightly lower than that measured during 2010-2011 at the site (7 $\mu g/m^3$) (Parworth et al., 2015) while for COR, the same value is reported in Fast et al. (2024) for the same campaign. In contrast, the lowest mass concentrations are observed at marine sites, ASI and ENA, with mean values of 0.96 and 0.54 $\mu g/m^3$, respectively. The mountain site GUC exhibits an intermediate concentration of 1.57 $\mu g/m^3$. These mean values are consistent with previous studies reporting $PM_1$ levels below 1 $\mu g/m^3$ in remote and pristine marine environments over the Pacific, Atlantic, and polar oceans (Zhou et al., 2023), as well as with

observations from high-altitude mountain sites where lower aerosol mass concentrations are typically found due to reduced anthropogenic influence (e.g., Fröhlich et al., 2015; Jimenez et al., 2009). It is important to note that the aerosol chemical composition exhibits strong seasonal variability, and the values presented here reflect specific measurement periods rather than long-term, annual averages, except at SGP, where long-term measurements are available.

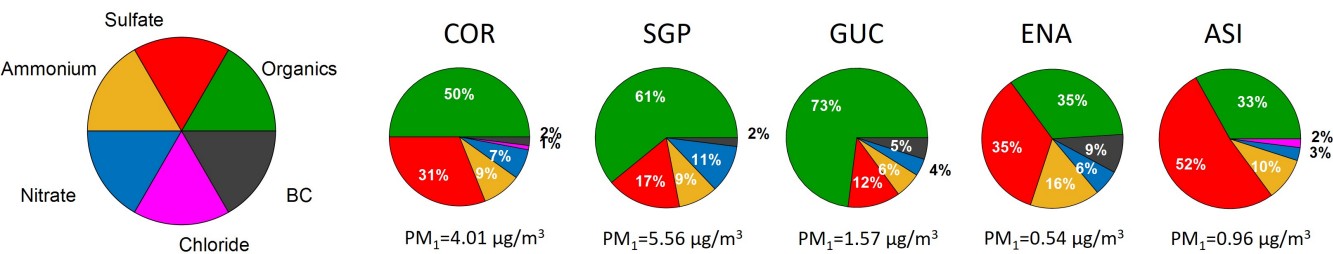

**Figure 3.** Pie chart of $PM_1$ mass concentration (OA, $SO_4^{-2}$, $NO_3^-$, $NH_4^+$, $Cl^-$ and BC) averaged for all the sites. Total mean $PM_1$ mass concentration for each site included.

For non-marine sites, the most abundant aerosol component is organic aerosol (OA) while at marine locations (ASI and ENA)

sulfate dominates. Among non-marine sites, the relative contribution of OA ranges from 50% at COR to 73% at GUC. The OA concentration is highest at SGP (2.30 $\mu g/m^3$) followed by COR (2 $\mu g/m^3$), while the OA concentration at marine sites





is less than 0.5 $\mu g/m^3$. Sulfate is the main contributor to aerosol mass in ASI, accounting for 52% of PM$_1$ (0.5 $\mu g/m^3$). At ENA, sulfate and organic have the same concentration values (0.19 $\mu g/m^3$), representing 35% each of the total PM$_1$ mass. The presence of sulfate at these two sites is likely mainly associated with sea salt particles (Lin et al., 2022), consistent with their

location in the marine environment. For COR, SGP, and GUC, sulfate is the primary inorganic component, with contributions of 31% at COR, 17% in SGP, and 12% in GUC. The high contribution of $SO_4^{2-}$ in COR has been linked to $SO_2$ emissions from small fires occurring outside Patagonia and the Atacama Desert (Fast et al., 2024).

The ammonium contribution ranges from 6% at the GUC mountain site (0.10 $\mu g/m^3$) to 16% at the ENA marine site (0.009 $\mu g/m^3$). At the continental sites, COR and SGP, ammonium accounts for 9% of the PM$_1$ mass concentration (0.36 and 0.50

$\mu g/m^3$, respectively). Marine sites exhibit slightly higher variability, with contributions of 16% at ENA and 10% at ASI. These differences reflect both emission sources and total aerosol load. In continental environments, higher ammonium concentrations are driven by local and regional anthropogenic sources, including agriculture (especially livestock and fertilizer use), road traffic, industrial activities, landfills, coal combustion, and biomass burning (Anderson et al., 2003; Sutton et al., 2000). In contrast, the lower total PM$_1$ mass concentration observed for marine environments leads to a higher relative contribution

of ammonium, despite low absolute concentrations. The ocean is one source for this ammonium (e.g., Quinn et al., 1988). Regional transport and secondary formation processes further enhance ammonium levels through the production of compounds such as ammonium sulfate and nitrate (Kang et al., 2018).

At most stations, nitrate plays a minor role (contribution less than 5%) except for the continental stations (SGP; 11% and COR; 7%). SGP shows the higher mean $NO_3^-$ concentration (0.6 $\mu g/m^3$), followed by COR (0.3 $\mu g/m^3$). The higher contribu-

tion of nitrate at continental sites is associated with anthropogenic emission sources such as fossil fuel combustion, biofuel combustion, and agricultural fertilization (Jaegle et al., 2005).

Among BC concentrations, the highest contributions are observed at ENA (9%; 0.05 $\mu g/m^3$), likely influenced by local human activity near the station, which is located within half a kilometer of the local airport (Wilbourn et al., 2024). At the mountain site GUC, BC concentrations remain low (0.42 $\mu g/m^3$), yet it accounts for 5% of PM$_1$ mass. At continental sites, BC contributes

less than 2% with concentrations of 0.11 $\mu g/m^3$ at SGP and 0.08 $\mu g/m^3$ at COR.

### 3.2.2 Composition-derived hygroscopicity, $\kappa_{\text{chem}}$

The bulk chemical composition is used to estimate the overall $\kappa_{\text{chem}}$ for each site, as explained in Section 2.4. In this study, $\kappa_{\text{chem}}$ is derived based on three variations of Equation 1: (i) including BC (Scheme 1); (ii) excluding BC (Scheme 2); and (iii) assuming a fixed $\kappa_{\text{chem}}$ of 0.3 for all aerosols (Scheme 3). Figure 4a shows the resulting $\kappa_{\text{chem}}$ values for each scheme at

sites with available chemical composition measurements. Scheme 1 could not be applied at ASI due to the dataset RH<40% constraint resulting in no harmonized absorption data at the site (Andrews et al., 2025a). Scheme 3, which assumes a constant value $\kappa_{\text{chem}}$ regardless of site characteristics, is represented as a horizontal line at all stations. Among all sites and for both Schemes 1 and 2, the marine stations (ENA and ASI) have the highest $\kappa_{chem}$ values (around 0.45), followed by the continental sites (COR and SGP, approximately 0.3), and the mountain site (GUC, around 0.23). In this context, applying a fixed value





of $\kappa_{\text{chem}} = 0.3$ (Scheme 3) tends to underestimate aerosol hygroscopicity in marine environments and overestimate it at the mountain site, while for the continental stations it provides a reasonably accurate approximation. The inclusion of BC in Scheme 1 results in slightly lower $\kappa_{\text{chem}}$ values compared to Scheme 2 across all sites, since BC is assumed to be completely hydrophobic ($\kappa_{BC} = 0$), thereby reducing the volume-weighted contribution of hygroscopic species. It is also worth noting that at marine sites, $\kappa_{chem}$ may be underestimated due to the inability of the ACSM to detect refractory sea salt, which can

significantly contribute to aerosol hygroscopicity in those regions (Deshmukh et al., 2025).

In general, $\kappa_{CCN}$ is lower than $\kappa_{chem}$ for all sites except ASI. Note that these two parameters cannot be directly compared since $\kappa_{CCN}$ only accounts for activated particles in the CCNC and its calculation depends primarily on the dry aerosol size distribution and CCN concentrations as a function of SS, while $\kappa_{chem}$ is based on chemical composition and its calculation here represents the aerosol particles in the size range sampled by the ACSM (40-1000 nm) (Watson, 2017). Depending on

the SS, the CCNC and ACSM may be measuring particles in different size ranges and with different compositions. Despite the methodological differences, the general trend is similar: continental and mountain sites show lower hygroscopicity values, while marine sites are characterized by higher hygroscopicity parameters. The higher $\kappa_{CCN}$ compared to $\kappa_{chem}$ obtained for ASI could be explained by the low $D_{crit}$ value observed at this site, which is near the lower end of the sampling interval of the ACSM.

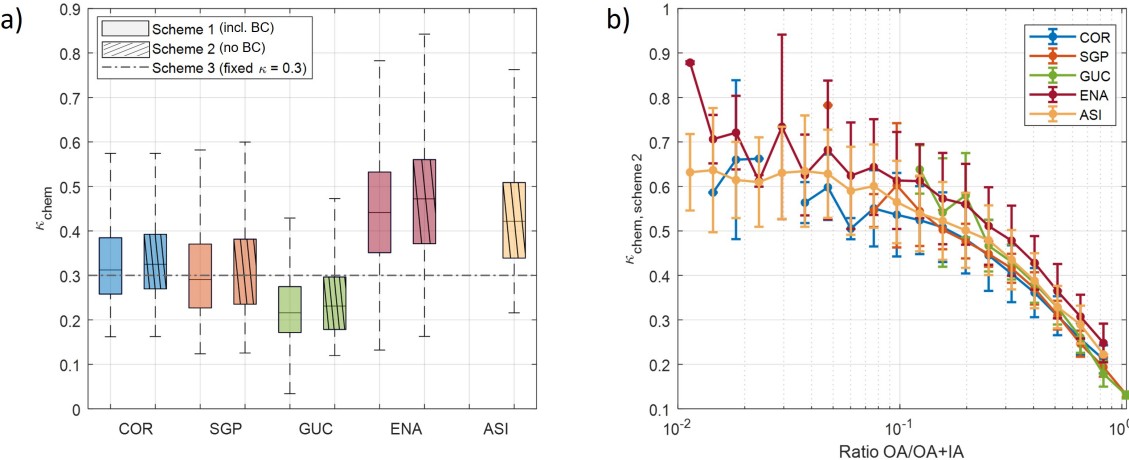

**Figure 4.** (a) Boxplots of $\kappa_{\text{chem}}$ values for Schemes 1 and 2 at all sites with available chemical composition measurements. The line inside each box indicates the median, the bottom and top edges of the box represent the 25th and 75th percentiles, and the whiskers extend from the ends of the interquartile range (IQR) to the most extreme data points within 1.5 times the IQR. Scheme 1 could not be applied at ASI due to the absence of harmonized BC measurements. Scheme 3, which assumes a constant $\kappa_{\text{chem}} = 0.3$, is represented as a horizontal line across all sites. (b) Relationship of the composition-derived $\kappa_{chem}$ from Scheme 2 to the binned and averaged ratio of organic (OA) to total (OA+IA) aerosol components. The vertical bars denote the standard deviation.

Figure 4b shows the variation in the chemical composition derived hygroscopicity parameter ($\kappa_{chem}$) from Scheme 2 as a function of the binned and averaged ratio of organic to total aerosol mass concentration (OA / [OA + IA]) for the five locations





with ACSM measurements. The data were binned into 30 logarithmically spaced intervals between 0.01 and 10. The standard deviation is represented for each averaged value. Figure S3 in the Supplement provides the corresponding analysis using Scheme 1 (ASI can not be included in this case due to the lack of harmonized BC measurements). For both schemes, a

clear decreasing trend in $\kappa_{chem}$ with increasing organic fraction is observed at all sites, reflecting that a higher contribution of organic aerosols reduces the overall hygroscopicity of the aerosol population. This behavior is consistent with the typically lower hygroscopicity of organic compounds relative to inorganic salts (Pöhlker et al., 2023). At low (OA / [OA + IA]) ratios (<0.1) $\kappa_{chem}$ becomes more noisy due to the lower number of data points, but appears to plateau between 0.5 and 0.7. When OA / [OA + IA] < 0.1, the volume fractions $\epsilon_i$ of sulfate, ammonium, and nitrate dominate, as these are the main inorganic

species at all sites (as shown in Fig. 3). Consequently, these species govern the sum in Eq. 1, and $\kappa_{chem}$ plateaus at their volume-fraction-weighted average value (approximately 0.5–0.7; see Table S4).

This pattern is further supported by the results presented in Figures 3 and 4a. GUC, the site with the highest organic fraction (73%), exhibits the lowest $\kappa_{chem,Sch2}$ value among all the sites ($\sim 0.2$). Similarly, the other two continental sites, SGP and COR, have intermediate OA fractions (61% and 50%, respectively) and correspondingly low $\kappa_{chem,Sch2}$ values ($\sim 0.25$ and $\sim 30$). In

contrast, the marine site ENA, with a lower organic fraction of 35%, presents a more balanced chemical composition—35% organics, 35% sulfate, and 16% ammonium—and a higher $\kappa_{chem,Sch2}$ ($\sim 0.47$). ASI, characterized by a dominant sulfate contribution (52%) and the lowest organic fraction among the sites (33%), exhibits a similar $\kappa_{chem,Sch2}$ to ENA ($\sim 0.45$). These results suggest that the organic fraction is a key driver of particle hygroscopicity, modulating the ability of the aerosol to take up water, thereby impacting the overall particle hygroscopicity (Aklilu et al., 2006; Dusek et al., 2010). In general, increasing organic

fraction leads to a reduction in $\kappa_{chem}$, while a higher contribution of inorganic species - particularly sulfate and ammonium - increases overall hygroscopicity (Petters and Kreidenweis, 2007).

### 3.2.3 CCN prediction using $\kappa_{chem}$

Using the calculated $\kappa_{chem}$ values, $N_{CCN}$ is estimated using $\kappa$-Köhler theory (Section 2.6.1). The predictions are made considering the three $\kappa_{chem}$ schemes. Figure 5 compares the predicted and measured CCN concentrations at all SS for the four sites

where all three schemes can be applied to allow fair evaluation of the performance of each approach. Note that ASI is excluded due to the lack of harmonized BC measurements at that station.





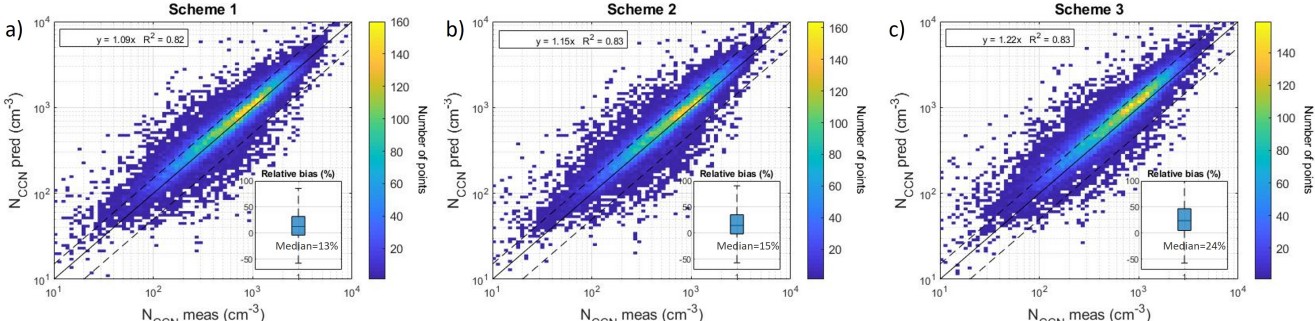

**Figure 5.** Log-log scatter plot of predicted CCN concentrations ($N_{CCN}$ pred) with respect to the observed CCN concentrations ($N_{CCN}$ meas) for all SS for all the sites except ASI using the three prediction schemes. A boxplot showing the relative bias is included- the central line represents the median, the box edges correspond to the 25th and 75th percentiles, and the whiskers extend from the ends of the interquartile range (IQR) to the most extreme data points within 1.5 times the IQR. Plots correspond to (a) Scheme 1 ($\kappa_{chem,Sch1}$), (b) Scheme 2 ($\kappa_{chem,Sch2}$) and (c) Scheme 3 (fixed $\kappa_{chem}$). The solid black line represents the 1:1 line and the dashed lines are the +/-50%.

Among the three schemes, the coefficient of determination ($R^2$) is virtually identical (0.82 or 0.83), indicating a similarly strong correlation between predicted and observed CCN concentrations for all schemes. Scheme 1 (Fig. 5a) shows the best overall agreement with observations, with a slope of 1.09 and the lowest median relative bias (13%), indicating a slight overall overprediction. Scheme 2 (Fig. 5b) shows a slightly higher slope of 1.15 and a median relative bias of 15%, reflecting a slightly higher overprediction compared to observations. However, the overall performance remains comparable to Scheme 1, with similar predictive capability despite not considering BC. Scheme 3 (Fig. 5c), which uses a fixed $\kappa_{chem}$, exhibits the highest slope (1.22) and the highest median relative bias (24%), pointing to a consistent tendency to overpredict $N_{CCN}$. We must consider the effect of the differences in the size ranges of the CCNC and the ACSM. While the CCNC has no lower size cutoff, the ACSM measures particles in the 40–1000 nm size range (Watson et al., 2018), which could lead to an underestimation of the predicted CCN concentrations if $D_{crit}$ is smaller than the ACSM lower size cutoff. However, such small $D_{crit}$ values are rare: the 10th percentile drops below 40 nm only at ENA (32–33 nm for SS $\geq$ 0.8%) and at ASI (18–28 nm for SS $\geq$ 0.4%). Therefore, the ACSM lower size cutoff may cause a slight underestimation of CCN at ASI and, to a lesser extent, at ENA, but provides comparable estimates at the other sites.

Figure S4 in the Supplemental provides further insight into the performance of each scheme across different stations by showing the $R^2$ and median relative bias (MRB) values per site—here, all available measurements for each scheme are included, and ASI is also considered for Schemes 2 and 3. Table S5 lists the number of data points available per site for each scheme. Continental stations (SGP, COR, GUC) show a slight overestimation of CCN concentration in all three schemes (MRB>0). This may be due to the presence of lower activity particles not fully accounted for by a bulk $\kappa_{chem}$ value. Nevertheless, the $R^2$ values remain high (between 0.77 and 0.82), indicating generally good predictive skill. At the marine station ENA, CCN concentrations are slightly overestimated, by a bit more (10-15%) than at continental sites. Still, $R^2$ values remain above 0.78 across all schemes, indicating robust predictive performance despite some variability in chemical composition. In contrast, an underestimation of CCN concentration is observed at the marine station ASI in Schemes 2 and 3 (MRB<0) (Scheme 1 cannot



be applied at this site). Although ASI exhibits high sulfate levels (52%) and relatively low organics (33%), as shown in Fig. 3,
sea salt — typically abundant and highly hygroscopic in marine environments — cannot be detected due to the limitations of
the ACSM. Consequently, the predictions calculated using Scheme 2 may underestimate the actual aerosol hygroscopicity at
this site. Similarly, in Scheme 3, the use of a fixed $\kappa_{chem}$=0.3 may underestimate the actual bulk aerosol hygroscopicity at ASI,
which likely exceeds this value due to the dominance of sulfate and the potential presence of unmeasured sea salt. Despite these
limitations, ASI shows the best agreement between predicted and measured CCN, with $R^2$ values of 0.97 and MRB<25% for
both schemes, suggesting the prediction framework performs well, possibly due to the relatively stable atmospheric conditions
and the less variable aerosol composition typical of remote marine environments (Saliba et al. (2020); Zuidema et al. (2015)).
An additional factor contributing to the $N_{CCN}$ underestimation at ASI could be that the median $D_{crit}$ at 0.8 and 1% SS is below
the ACSM detection limit (40 nm). Consequently, some particles activated as CCN are not captured in the chemically derived
predictions, leading to measured CCN concentrations exceeding the predicted values. Although ENA is also a marine station,
its higher organic fraction (35%) likely reduces the influence of unmeasured sea-salt particles — which are more hygroscopic
— on $\kappa_{chem}$, resulting in an overestimation of $N_{CCN}$. Schmale et al. (2018) reported a consistent overestimation of predicted
CCN concentrations using different $\kappa_{chem}$ schemes at 6 measurement sites (only one of the seven sites studied in Schmale et al.
(2018) underestimated CCN measurements, and it was also a marine site). Although the use of composition-derived values
$\kappa_{chem}$ consistently reduces bias and tightens the fit to measured CCN, our results are consistent with those of Schmale et al.
(2018) suggesting that even a constant bulk $\kappa_{chem}$ = 0.3 provides a realistic first-order estimate of CCN number concentrations
in diverse environments.

## 3.3 Aerosol optical properties and CCN prediction

### 3.3.1 Overview of aerosol optical properties

Aerosol optical measurements are available at 7 of the 10 sites (not available for SBS-CP, SBS-SPL and ASI). Figure 6 provides
an overview of key aerosol optical parameters for all sites, including $\sigma_{sp}$ and $\sigma_{ap}$, and four derived parameters: BSF, SAE,
AAE and SSA. All measurements used in this analysis correspond to $PM_{10}$ aerosol size cut hourly data and are reported at 550
nm, or for the blue/red wavelength pair for SAE and AAE. As filtering criteria, for the calculation of the derived parameters,
measurements with $\sigma_{sp} < 0.5$ $Mm^{-1}$ were not considered and unphysical values were also excluded, i.e., SSA and BSF outside
0–1. In addition, negative SAE and AAE values were also excluded. On average, the combined constraints eliminated about
4% of the data across all stations, although at MOS up to 17% of the measurements were discarded. The filter responsible for
most exclusions varied depending on the station, while the SSA constraint was generally the least restrictive, removing the
fewest data points. It is important to note that the values presented here correspond to specific measurement periods rather than
year-round averages, except for SGP and GUC, where more than 1 year of AOP observations are available and allow for a
more representative characterization.





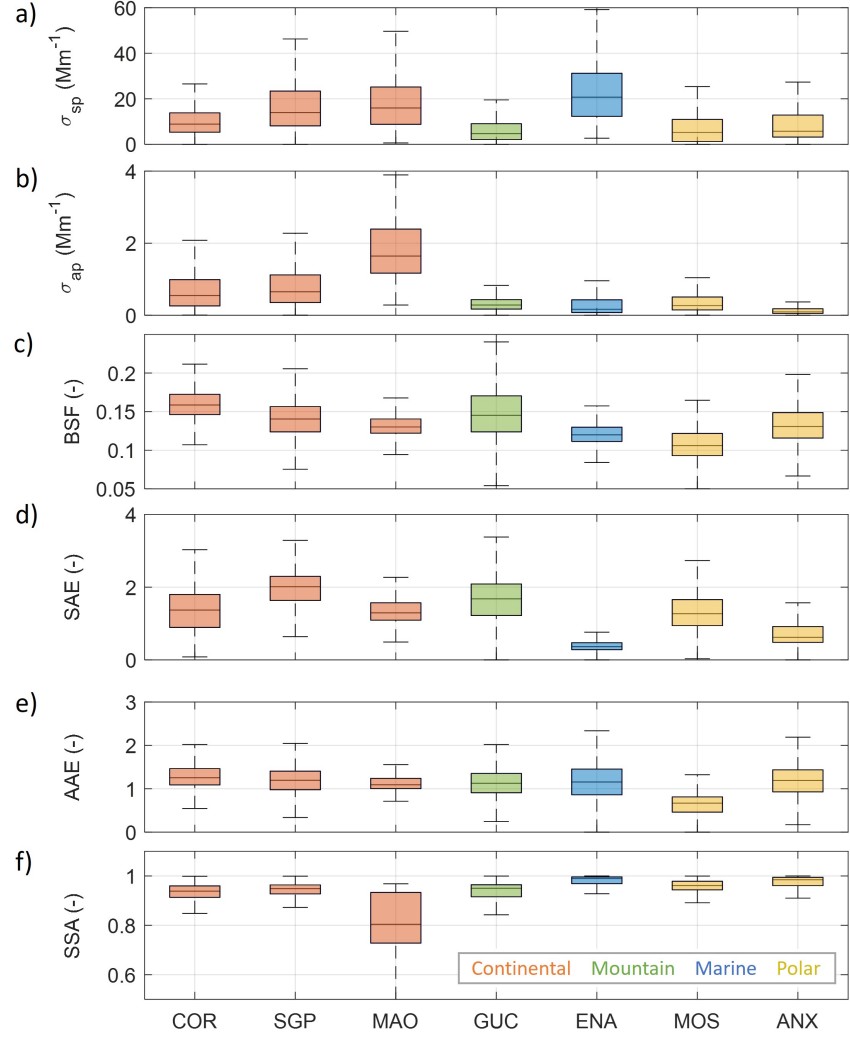

**Figure 6.** Boxplots of the distribution of aerosol optical properties at all sites. (a) $\sigma_{sp}$, (b) $\sigma_{abs}$, (c) $BSF$, (d) $SAE$, (e) $AAE$ and (f) $SSA$. Median values (black lines), 25th–75th percentiles (black boxes) and the whiskers extend from the ends of the interquartile range (IQR) to the most extreme data points within 1.5 times the IQR.

The scattering coefficient (Fig. 6a) shows notable variability across sites, reflecting differences in aerosol loading. The highest median $\sigma_{sp}$ is observed at the marine site ENA (e.g., 20.7 Mm$^{-1}$), which contrasts with the low PM$_1$ concentration at this site. This is likely due to high concentrations of supermicron sea salt particles commonly found in marine-influenced environments (Vaishya et al., 2011). This site is followed by MAO, SGP, and COR continental stations, with median values of 15.9, 13.9, and 8.9 Mm$^{-1}$, respectively. In contrast, the mountain site GUC and the polar locations (MOS and ANX) show the lowest

median scattering coefficients (e.g., 4.7, 5.2, and 5.7 Mm$^{-1}$, respectively), consistent with their remote and cleaner atmospheric




conditions. These findings align with those reported by Laj et al. (2020), where values below 10 Mm$^{-1}$ were observed for polar environments and mountain sites.

The absorption coefficient (Fig. 6b) has a different pattern at the sites than the scattering coefficient. The highest median $\sigma_{\mathrm{ap}}$ is observed at the continental site MAO (1.63 Mm$^{-1}$), suggesting a strong presence of absorbing particles, likely from biomass

burning and anthropogenic emissions (Rizzo et al., 2013). This is followed by the other continental stations, COR and SGP, with median values of 0.55 and 0.65 Mm$^{-1}$, respectively. Marine and polar sites exhibit significantly lower values, with ENA, MOS and ANX showing median concentrations of 0.17, 0.27, and 0.09 Mm$^{-1}$. The mountain site GUC reports a moderate absorption level of 0.28 Mm$^{-1}$, in line with previous findings for high-altitude, remote locations, where aerosol absorption tends to be limited due to the absence of nearby combustion sources (Collaud Coen et al., 2018).

The back-scattered fraction (Fig. 6c), which is a proxy for particle size in the aerosol population, shows the highest median values at continental and mountain sites. The highest BSF is observed at COR (0.16), followed by SGP, GUC, and MAO, all with median values of 0.14. These elevated BSF values indicate a greater contribution from smaller particles. Marine and polar sites (ENA, ANX, and MOS) show smaller median BSF values in the range 0.10–0.13. This highlights the different source regimes - sea spray and remote transport in the marine boundary layer, and aged background aerosol in polar regions.

The scattering Ångström exponent (Fig. 6d) provides complementary information to BSF, as it is more sensitive to particles in the upper accumulation and coarse modes (Collaud Coen et al., 2007). The highest SAE values are observed at continental and mountain sites such as SGP (2.01), GUC (1.67), and COR (1.37), consistent with the prevalence of fine-mode aerosols from anthropogenic and biomass burning sources. At COR, frequent dust transport during the austral spring may explain its relatively lower SAE compared to other continental sites (Varble et al., 2019). In contrast, lower SAE values at marine and

polar sites—ENA (0.36), ANX (0.62), and MOS (1.27) — suggest a stronger influence of coarse-mode particles such as sea spray or aged background aerosol.

The absorption Ångström exponent (Fig. 6e), which describes the wavelength dependence of aerosol light absorption and provides insight into aerosol composition, shows relatively consistent median values across most sites, ranging between 1.1 and 1.3, but with the higher percentiles ranging up to 2 - 2.5. The median values reflect locations with absorption primarily

due to BC based on the Cappa et al. (2016) AAE/SAE matrix, while the higher AAE values indicate occasional incursions of absorbing aerosols related to dust or biomass burning organics (Cazorla et al., 2013; Kirchstetter et al., 2004). In contrast, the polar site MOS exhibits a notably lower median AAE of 0.67. AAE values below 1 have been previously reported at remote Arctic and marine sites (Schmeisser et al., 2018), although such low AAE values may also be partially influenced by measurement artifacts in the presence of coarse-mode aerosols (Bond et al., 1999).

Finally, the single scattering albedo (Fig. 6f), which indicates the relative contribution of absorbing particles to aerosol extinction coefficient, shows high values across most sites (>0.9), suggesting the dominance of scattering aerosols. ANX, MOS, and ENA, which are all marine influenced, have median SSA > 0.95, while GUC, SGP and COR have median SSA values closer





to 0.9. The lowest median SSA is found at MAO (0.80), indicating a relatively more absorbing aerosol mixture at this site consistent with anthropogenic and biomass sources.

### 3.3.2 CCN predictions using aerosol optical properties (S2019)

Following the S2019 methodology described in Section 2.6.2, Figure 7 compares predicted CCN concentrations using (a) the original S2019 equation ($N_{CCN,S2019}$) and (b) the new version of the S2019 equation derived using the original data of S2019 and the data from the stations in this study ($N_{CCN,new}$), against measured CCN concentrations ($N_{CCN}$ meas) for the seven sites with optical properties in this study and for all SS. The number of data points for each site used in the comparison—identical for both equations—is provided in Table S5 in the Supplemental. The comparison shows an increase in the regression slope from 0.72 in plot (a) to 0.86 in plot (b), indicating a better agreement between predicted and measured $N_{CCN}$ when using the new equation. The coefficient of determination ($R^2$) remains unchanged (0.61), suggesting that the overall model performance is comparable in terms of explained variance. The median relative bias decreases in absolute value from –27% in (a) to –8% in (b) as the number of sites increases, indicating a reduced underestimation in the predictions. Meanwhile, the similar length of the MRB whiskers in both cases suggests that the variability remains comparable, even when a broader range of stations and aerosol conditions are included. However, the interquartile range decreases from 81 to 69, indicating reduced variability in errors. This reduction in MRB, together with the smaller IQR, reflects an improvement in prediction accuracy, with fewer extreme deviations and a more balanced distribution of errors. Consequently, the new equation provides CCN predictions that are more reliable and closely aligned with the measured CCN concentrations across the full range of conditions.

Figure S5 in the Supplemental provides additional insight into the performance of both equations across different stations by displaying the site-specific $R^2$ and MRB (median relative bias) values. As observed in Fig. 7, the coefficients of determination remain largely unchanged between the two equations. For continental (COR, SGP, MAO) and mountain (GUC) sites, CCN concentrations tend to be slightly underpredicted with MRB<0 (Fig. S5a), whereas overpredictions are more common at marine (ENA) and polar (MOS, ANX) sites (MRB>0; Fig. S5a). The new equation (Fig. S5b) generally increases the predicted $N_{CCN}$ values, leading to an overall improvement in prediction accuracy. Figure S6 shows the slope and relative bias for each measured SS between the predicted and the measured CCN concentrations considering the new equation. Excluding the lowest SS (0.1%), both the slope and the median relative bias remain relatively stable across all SS values, indicating that the predictive equation performs consistently well regardless of SS. The larger deviation observed at 0.1% SS may be attributed to the logarithmic function used to capture the dependence of $N_{CCN}$ on SS. These results confirm that the original S2019 equation performs well across a wide range of conditions, even when evaluated with an extended dataset. However, the new equation proposed in this work provides a more accurate and balanced estimation of $N_{CCN}$, particularly by reducing systematic underestimation and improving agreement across the full concentration range.



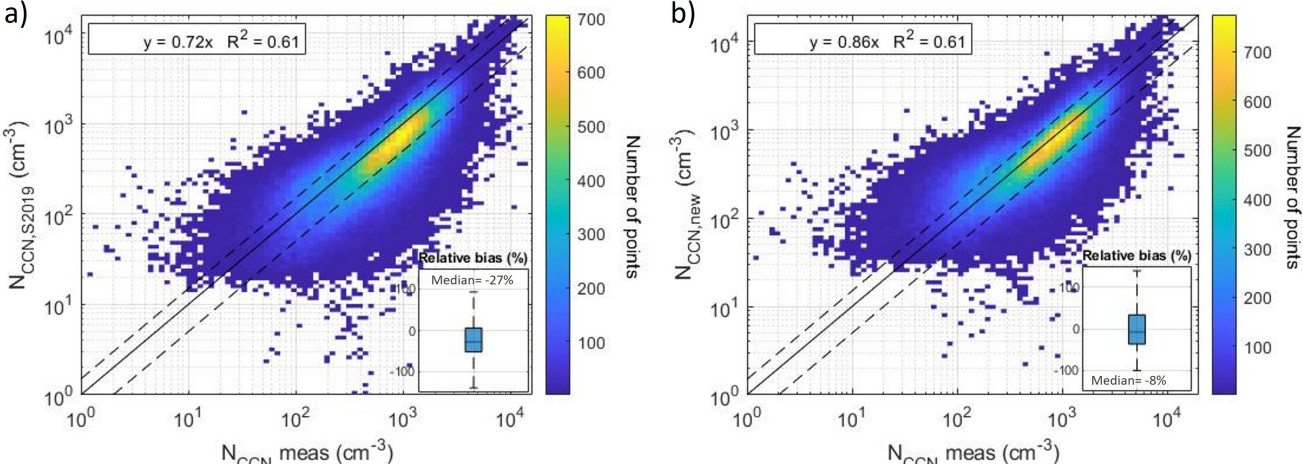

**Figure 7.** Log-log scatter plot of predicted CCN concentrations with respect to the observed CCN concentrations ($N_{CCN}$ meas) considering (a) equation in S2019 ($N_{CCN,S2019}$) and (b) new equation ($N_{CCN,new}$; based on 13 sites). The data plotted is only for the seven sites with optical data in this study (i.e., sites shown in Fig. 6). The solid black line represents the 1:1 line and the dashed lines are the +/-50%. A boxplot showing the relative bias is included. The boxes represent the interquartile range (25th–75th percentiles), with black lines indicating the median values and whiskers extending from the ends of the interquartile range (IQR) to the most extreme data points within 1.5 times the IQR.

### 3.3.3   CCN prediction with random forest model using optical properties

To further explore the potential of aerosol optical properties to predict CCN concentrations, a random forest model was implemented to estimate the $C$ and $k$ parameters of the Twomey equation. As input variables for the RF model, the same set of AOPs as in the S2019 equation (Section 3.3.2) is considered: $\sigma_{sp}$, BSF and SAE. Once the model is run, the predicted parameters are used to compute CCN concentrations across a range of SS. The performance of the model is evaluated by comparing these predictions based on RF with measured CCN values, allowing a direct comparison with the results of the S2019

parameterizations.

Figures 8 and S7 present the results of the RF model. Figures 8 (a) and (b) display the relative importance of each input variable in predicting the $C$ and $k$ parameters, respectively, while Figure S7 compares the observed and RF-predicted $C$ and $k$ parameters. For the $C$ parameter, $\sigma_{sp}$ contributes the most, followed by BSF and SAE, highlighting the dominant role of the total particle loading in determining the potential CCN concentration. In contrast, BSF is the most important variable in $k$

prediction, followed by SAE and $\sigma_{sp}$, suggesting that the physicochemical properties of the particles, more strongly reflected by BSF and SAE, are more relevant to capture the chemical sensitivity embedded in $k$. These results are consistent with previous studies that have shown that $C$ is primarily influenced by aerosol number concentration and total mass loading, while $k$ reflects aerosol hygroscopicity and size distribution (Cohard et al., 1998; Jefferson, 2010; Vié et al., 2016; Rejano et al., 2021). Typically, high $C$ values are found under polluted conditions with high particle number concentrations, whereas low $k$

values are associated with particles exhibiting higher hygroscopicity and larger sizes (Martins et al., 2009; Pöhlker et al., 2016;





Jayachandran et al., 2020). Thus, independent prediction of these two parameters offers valuable information on the abundance and physicochemical properties of aerosols that influence CCN activation.

Figure 8(c) shows the comparison of the predicted CCN concentrations, calculated using the RF-derived $C$ and $k$ values, and measured CCN concentrations across all supersaturations. The result shows a slope of 0.90 and a $R^2$ of 0.62, indicating good

agreement between predictions and measurements. The inset boxplot shows the distribution of relative bias, with a median value of approximately 19%, indicating a overall overestimation.

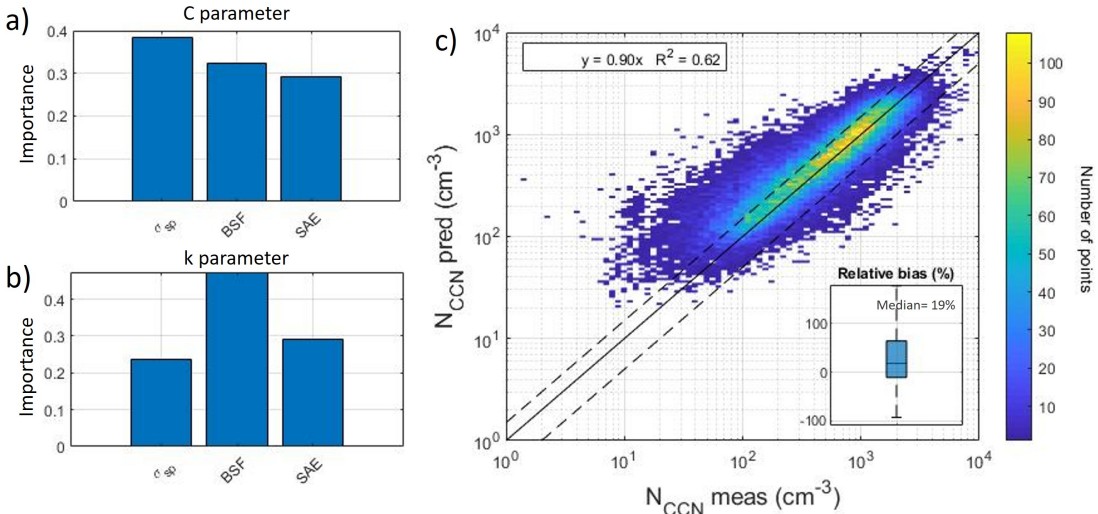

**Figure 8.** Importance of input variables in the random forest model considering AOPs used in S2019 ($\sigma_{sp}$, BSF, and SAE) for (a) $C$ and (b) $k$ parameters. (c) Log-log scatter plot of predicted CCN concentrations ($N_{CCN}$ pred) versus observed concentrations ($N_{CCN}$ meas) using a RF model to estimate the parameters of the Twomey equation. The solid black line represents the 1:1 line and the dashed lines are the +/-50%. A boxplot showing the relative bias is included. Boxes show the interquartile range (IQR, 25th–75th percentiles), with black lines indicating median values, and whiskers extending from the ends of the IQR to the most extreme data points within 1.5 times the IQR.

RF models can take advantage of additional informative features without a significant loss in predictive performance (Breiman, 2001) so, as the next step, the RF model is extended by including the full set of AOPs as predictors: $\sigma_{sp}$, BSF, SAE, $\sigma_{ap}$, AAE and SSA. Although some of these variables are strongly correlated (see Fig. S8), RF models are known to be robust to

multicollinearity (Gregorutti et al., 2017). Figure 9c compares the predicted CCN concentrations—calculated using RF-derived $C$ and $k$ values from the full AOP set—with the observed values. The extended model achieves a slope of 0.91 and an $R^2$ of 0.69, slightly improving upon the performance of the RF model using only the three Shen-based variables (slope = 0.90, $R^2$ = 0.62). The median relative bias also decreases slightly from 19% (three-variable case) to 15% (full AOP set), with comparable interquartile ranges (–92 to 180 vs. –88 to 145). To assess the RF models' performance across different SS levels, Figure S9

presents the slope and median relative bias for both schemes. Results are consistent across the SS range, with slopes ranging from 0.80 to 0.99 and median relative biases between 8% and 32%, indicating that the predictive capability of the RF models is independent of SS. Finally, Figure S10 in the Supplemental Material shows site-specific $R^2$ values comparing predicted and





measured CCN concentrations for both RF schemes-the S2019 AOPs (Fig. S10a) and the full AOP set (Fig. S10b). While the overall performance is similar, the inclusion of all AOPs—despite some strong inter-variable correlations (Fig. S8)—slightly improves both the coefficient of determination and the bias across all sites, supporting a more accurate prediction of CCN concentrations.

To better understand the source of these improvements in CCN prediction, we next analyze the relative importance of the input variables used to estimate the $C$ and $k$ parameters when using the full AOPs set. Figures 9 (a) and (b) display the relative importance of each input variable in predicting the $C$ and $k$ parameters, respectively, while plots in Fig. S11 compare the observed and RF-predicted $C$ and $k$ parameters. AAE is identified as the most important input for the prediction of $k$ (Fig. 9b), followed by SAE and BSF, suggesting that the chemical sensitivity embedded in $k$ is better captured when accounting for absorption-related properties. For the prediction of the $C$ parameter, BSF is the most important variable (Fig. 9a), followed by SAE and AAE, while $\sigma_{sp}$ is of relatively lower importance. This result contrasts with the previous model (Fig. 8a), where $\sigma_{sp}$ dominated, highlighting that including absorption-related parameters redistribute the contribution across variables. As previously mentioned, some of these variables are strongly correlated (Fig. S8) and the model tends to distribute the importance among correlated variables affecting overall predictive performance (Genuer et al., 2010).

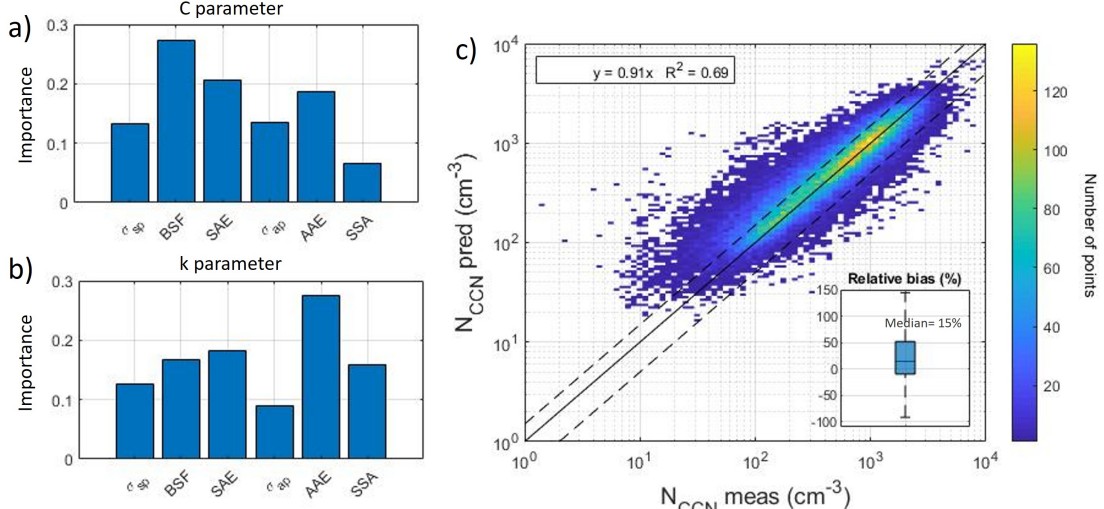

**Figure 9.** Importance of input variables in the random forest model considering all AOPs ($\sigma_{sp}$, BSF, and SAE, $\sigma_{ap}$, AAE, and SSA) for (a) $C$ and (b) $k$ parameters. (c) Log-log scatter plot of predicted CCN concentrations ($N_{CCN}$ pred) versus observed concentrations ($N_{CCN}$ meas) using a RF model to estimate the parameters of the Twomey equation. The solid black line represents the 1:1 line and the dashed lines are the +/-50%. A boxplot of the relative bias is included. Boxes show the interquartile range (IQR, 25th–75th percentiles), with black lines indicating median values, and whiskers extending from the ends of the IQR to the most extreme data points within 1.5 times the IQR.

To further analyze how different AOPs contribute to the prediction of the $C$ and $k$ parameters, Figure 10 presents heatmaps of variable importance for models using different combinations of AOP inputs for $C$ (Figure 10a) and $k$ (Figure 10b). In these heatmaps, each row corresponds to a model run (the first row includes all AOPs; subsequent rows exclude one AOP at a time),





and each column represents one of the six AOPs. Analyzing these heatmaps reveals that BSF remains the most important predictor of $C$, except when $\sigma_{\mathrm{sp}}$, $\sigma_{\mathrm{ap}}$, or BSF itself are excluded from the model. In these cases, the model shifts its reliance to a closely related variable: AAE becomes the dominant predictor when BSF is removed, while $\sigma_{\mathrm{ap}}$ and $\sigma_{\mathrm{sp}}$ substitute each other when one is absent. This behavior likely reflects the partial redundancy and strong interdependence among BSF, AAE, $\sigma_{\mathrm{sp}}$, and $\sigma_{\mathrm{ap}}$. Indeed, their relationships are supported by the Spearman correlation coefficients (Fig. S8 in the Supplemental): BSF and

$\sigma_{\mathrm{sp}}$ are negatively correlated ($\rho_s = -0.41$), $\sigma_{\mathrm{sp}}$ and $\sigma_{\mathrm{ap}}$ show a strong positive correlation ($\rho_s = 0.68$), and BSF and AAE are moderately correlated ($\rho_s = 0.36$). While these correlations help explain why certain variables gain importance when others are removed, it is important to note that RF variable importance also depends on how much each variable contributes to reducing prediction error across the ensemble, not solely on pairwise correlations (Breiman, 2001). For the prediction of $k$ (Figure 10b), the AAE is the most important predictor under the full model. Removing AAE shifts the top rank to BSF, again reflecting

their correlation. This result highlights the RF model's ability to reallocate predictive importance among partially redundant features, relying on combinations of variables that together best capture the relevant information rather than depending on any single one.

| C prediction | $\sigma_{sp}$ | $\sigma_{ap}$ | BSF | SAE | SSA | AAE |
|---|---|---|---|---|---|---|
| All AOPs | 0.13 | 0.13 | 0.28 | 0.21 | 0.06 | 0.19 |
| No $\sigma_{\mathrm{sp}}$ | | 0.34 | 0.20 | 0.19 | 0.12 | 0.15 |
| No $\sigma_{\mathrm{ap}}$ | 0.27 | | 0.17 | 0.22 | 0.17 | 0.17 |
| No BSF | 0.11 | 0.19 | | 0.28 | 0.08 | 0.34 |
| No SAE | 0.16 | 0.21 | 0.36 | | 0.07 | 0.20 |
| No SSA | 0.18 | 0.21 | 0.22 | 0.21 | | 0.18 |
| No AAE | 0.15 | 0.12 | 0.35 | 0.29 | 0.09 | |

(a)

| k prediction | $\sigma_{sp}$ | $\sigma_{ap}$ | BSF | SAE | SSA | AAE |
|---|---|---|---|---|---|---|
| All AOPs | 0.12 | 0.09 | 0.17 | 0.18 | 0.16 | 0.28 |
| No $\sigma_{\mathrm{sp}}$ | | 0.18 | 0.16 | 0.18 | 0.21 | 0.27 |
| No $\sigma_{\mathrm{ap}}$ | 0.18 | | 0.17 | 0.17 | 0.22 | 0.26 |
| No BSF | 0.11 | 0.10 | | 0.22 | 0.14 | 0.42 |
| No SAE | 0.13 | 0.11 | 0.21 | | 0.19 | 0.36 |
| No SSA | 0.20 | 0.18 | 0.17 | 0.19 | | 0.26 |
| No AAE | 0.11 | 0.11 | 0.35 | 0.27 | 0.16 | |

(b)

**Figure 10.** Heatmap of input variable importance in the Random Forest model for (a) $C$ and (b) $k$ parameters. Each row corresponds to a RF model in which one AOP has been removed, while each column represents the importance assigned to each available AOP in that model. The variable with the highest importance in each prediction is shown in red; importance values ≥0.20 are shown in orange; values between 0.15 and 0.19 in dark yellow; and values <0.15 in light yellow.

RF model results could be influenced by the differences in the availability of data at each measurement site, providing better results for those sites where datasets are longer. Therefore, to evaluate the influence of each location on model generalization

when considering all AOPs, a LOSO cross-validation approach is applied as explained in section 2.6.3. Figure S12 in the Supplement shows the variable importance for each site in the LOSO iteration. In each subplot, the name of the site excluded is indicated. The importance of predictors remains consistent across sites: AAE and SAE typically dominate the prediction of $k$, while BSF, SAE and AAE are more important for predicting $C$. This consistency confirms that no single site influences feature selection within the model. Notably, when SGP — the site with the largest number of observations — is excluded, some shifts

in variable importance are observed. However, these changes are not large enough to affect the overall importance, suggesting that the 70/30 approach used in the main analysis is not biased by the dominance of SGP data. Figure S13 in the Supplement shows the comparison between predicted and observed CCN concentrations at each excluded site. Slopes range from 0.38 in





ENA to 1.87 in MOS, and $R^2$ values from 0.03 to 0.56. Although predictive performance remains good for most sites, the model shows reduced accuracy at marine and polar locations (e.g., ENA, MOS, ANX). This is likely due to the fact that the training data are dominated by continental stations, limiting the model's ability to capture the distinct AOP characteristics of marine and polar environments.

A recently published study by Wang et al. (2025a) used an ensemble of multiple machine learning tools to investigate the ability of AOPs to predict CCN concentrations at 5 sites which are common to this study (SGP, GUC, ENA, ASI and MOS). As input variables, Wang et al. (2025a) uses $\sigma_{sp}$, BSF, SAE and SSA at the different wavelengths. The $R^2$ values obtained ranged between 0.2 to 0.63, depending on the predictive model construction. Their ensemble model was trained specifically for each site and for SS=0.4%, aiming at optimizing their predictive potential to the unique atmospheric conditions of each site. In our case, we decided to apply the RF model to the whole range of SS and to all sites together in order to provide a general model that performs reasonably well at most atmospheric conditions.

## 4 Discussion of CCN prediction methods

Direct measurements of CCN concentration are less common than other aerosol properties measurements. Multisite harmonization efforts combining CCN and other collocated aerosol measurements (e.g., Schmale et al., 2017; Andrews et al., 2025a) can strengthen global prediction frameworks. Reliable CCN predictions from commonly measured aerosol properties would offer a cost-effective and scalable alternative to direct measurements, expanding the scope of aerosol–cloud interaction studies.

Several methodologies for the prediction of CCN concentrations have been reported in the literature. Approaches based on aerosol chemical composition, as considered in this work, apply $\kappa$-Köhler theory to derive CCN activity from bulk or size-resolved chemical measurements (Gunthe et al., 2009; Jurányi et al., 2010; Wang et al., 2010), providing a physically grounded estimate that captures the influence of composition on particle activation. Optical property–based approaches use measured aerosol optical characteristics as empirical proxies for CCN concentrations (Ghan et al., 2006; Shinozuka et al., 2009; Liu and Li, 2014), offering a simple and cost-effective method, particularly when long-term observational datasets are available. Other methods rely on particle number size distributions (PNSD) combined with either a critical activation diameter or the aerosol hygroscopicity parameter $\kappa$ derived from hygroscopic growth measurements (Ervens et al., 2007; Cai et al., 2018), providing predictions that directly account for particle size and activation behavior. Parameterization schemes based on aerosol activation properties, such as size-resolved activation ratios and inferred critical diameters, have been evaluated in several field campaigns, demonstrating robust performance across diverse environments (Deng et al., 2013).

Among PNSD-based methods, the simplest assumes a single activation diameter from which all particles are activated, typically 50–150 nm depending on SS (Lihavainen et al., 2003). This approach has been used in other studies (Asmi et al., 2011; Kerminen et al., 2012; Rose et al., 2017; Rejano et al., 2024) providing satisfactory CCN estimations. This simple approach to estimate CCN from PNSD and $D_{crit}$ is included here to enhance the discussion. The $D_{crit}$ values assumed in this work—150, 110, 80, 65, 53, and 49 nm for SS=0.1, 0.2, 0.4, 0.6, 0.8, and 1.0%, respectively — correspond to the median $D_{crit}$ for each





SS, obtained from the median values across stations, and are in line with previous studies (Bougiatioti et al., 2011; Jurányi et al., 2011; Schmale et al., 2018). Figure S14 presents the results of the PNSD prediction method, showing predicted versus measured CCN concentrations across the 10 sites.

Figure 11 shows the MRB between predicted and measured CCN concentrations across all SS for all the methods tested in this study. Positive MRB values indicate overprediction and negative values underprediction. The simple $D_{crit}$ approach yields a

MRB of –5% (shown in green) indicating an excellent agreement with observations. We next compare the aerosol optical and chemical predictions to each other and to this $D_{crit}$ approach.

In this study we evaluated three chemistry-based prediction schemes: including BC (scheme 1), excluding BC (scheme 2), and assuming constant $\kappa_{chem} = 0.3$. The first two performed similarly (MRB < 15%), while scheme 3 exhibits higher overprediction (24%). Comparable correlations to the other two methods suggest that even a bulk $\kappa_{chem}$ value can provide a first-order CCN

estimate across diverse environments. The overprediction observed with this method is consistent with previous applications of this approach. Schmale et al. (2017) reported a general overprediction of different $\kappa_{chem}$ schemes at 7 sites, and similar results were found at a high-mountain site by Rejano et al. (2024). Two main limitations of the chemical prediction method contribute to this bias. First, it is based on bulk chemical composition measurements (size-resolved chemical composition measurements are rare) which assumes that particles are internally mixed and chemically homogeneous regardless of size (Wang et al., 2010;

Ren et al., 2018). Second, it requires assumptions about the chemical species present (e.g., sulfate forms, organic types) in the atmosphere, which can introduce large variability in the predictions (Schmale et al., 2018; Rejano et al., 2024).

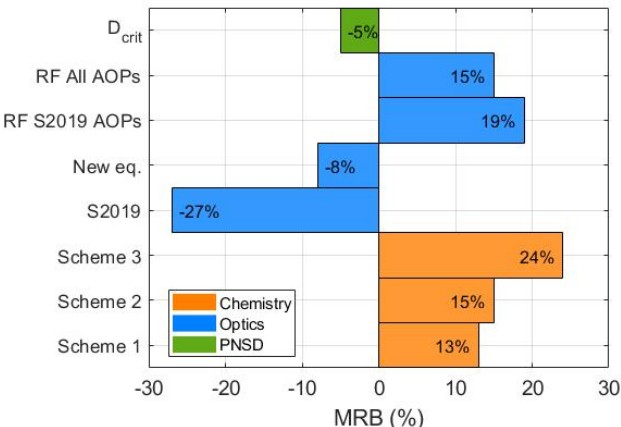

**Figure 11.** Median relative bias (MRB,%) between predicted and measured $N_{CCN}$ across all SS for different prediction methods. Each box represents a different predictive method (applied for the sites with available data as described in each section). Positive values indicate overprediction, while negative values indicate underprediction.

AOPs provide indirect information on particle size and composition and have been extensively measured for decades (Laj et al., 2020; Collaud Coen et al., 2020). Shen et al. (2019) developed an empirical parameterization to estimate CCN concentrations from three AOPs: $\sigma_{sp}$, BSF and SAE. In this study, we applied their method with additional sites, deriving new coefficients for





their empirical parameterization. As shown in Fig. 11, the original S2019 equation performs well across diverse conditions, but the updated version reduces underestimation (MRB) from –27% to –8% and achieves tighter agreement across the full $N_{CCN}$ range (blue bars labeled S2019 and New eq.). The new Shen-based parameters perform comparably to the $D_{crit}$ approach in terms of MRB and other metrics (e.g., slope and $R^2$). This S2019 approach is simple to apply and suitable where only nephelometer data are available, enhancing its applicability to global datasets. The historical availability of AOP data can

potentially facilitate broader spatial and temporal coverage of CCN estimates, albeit with more uncertainty than direct CCN observations or predictions based on aerosol size distributions.

Finally, we introduced a new approach based on the Twomey equation parameters, using a RF model to predict $C$ and $k$. Inputs included both the three AOPs from the S2019 equation ($\sigma_{sp}$, BSF, SAE) and the full set of available AOPs ($\sigma_{sp}$, BSF, SAE, $\sigma_{ap}$, AAE, SSA). As shown in Fig. 11, incorporating all AOPs reduced the MRB between predicted and measured $N_{CCN}$ from

19% (S2019 variables) to 15%, demonstrating the benefit of additional optical predictors. The MRB values are comparable to to scheme 1 and 2 of the chemical prediction, though not as low as those from the updated S2019 or $D_{crit}$ approaches. However, the RF analysis of AOP importance provides insights into the prediction of $C$ and $k$ that cannot be obtained with other prediction methods based on optical properties.

There are still many ways in which the CCN prediction schemes based on aerosol optical or chemical properties can be ex-

panded. In particular, a recent study (Wang et al., 2025b) shows that using dry scattering measurements instead of ambient-RH conditions results in a significant improvement of CCN concentration estimations—an error that increases with RH. This highlights a potential systematic bias in approaches relying solely on dry optical observations. Observational datasets such as those compiled by Burgos et al. (2019), with co-located scattering-related hygroscopicity, f(RH), at multiple sites, represent a key resource for future work. Leveraging such datasets could help refine CCN prediction models under ambient humidity and reduce

associated uncertainties.

Also, although this study combines information from 10 measurement sites, there are similar datasets at additional sites, that would be interesting to combine to have those additional co-located measurements harmonized. A potential application of the RF model and the new S2019 equation developed in this study is to look at long-term aerosol optical measurements to estimate CCN concentrations and expand the global and temporal coverage of CCN estimates. Additionally, those results could be used

to evaluate global models performance (Fanourgakis et al., 2019).

## 5   Conclusions

This work presents a comprehensive phenomenological study of in-situ aerosol microphysical, CCN activation, chemical composition, and optical properties at ten surface sites across diverse environments. Several CCN prediction methods using the chemical composition and aerosol optical properties were evaluated.

Analysis of aerosol microphysical properties and CCN activation at 0.4% SS reveals a wide variability between environments. The polar and marine sites exhibited the lowest concentrations of $N_{tot}$ and $N_{CCN}$, with values below 400 $cm^{-3}$ and 255





$cm^{-3}$, respectively. Despite similar particle concentrations at these remote sites, the significant variability in $D_{crit}$ and AF underscores the importance of size distribution and chemistry in CCN activation. In contrast, continental sites exhibited the highest $N_{tot}$ and $N_{CCN}$ (>2000 $cm^{-3}$ and 659 $cm^{-3}$, respectively) with fairly similar AF values (0.25-0.38) and a relatively
narrow range in $D_{crit}$ (76-98 nm). The mountain sites were more similar to the continental sites than the remote sites in terms of aerosol concentrations, but generally exhibited lower AF (<0.24).

The chemical composition analysis of the sites with ACSM measurements shows that organics dominate in continental and mountain sites (50–73% of $PM_1$), while marine stations are sulfate-rich (35–52% of $PM_1$). Total $PM_1$ mass ranges from 0.54 to 5.5 $\mu g/m^3$ across sites. Ammonium and nitrate reflect local emissions at the sites and BC is a minor fraction (<9%) of the
aerosol mass. A $\kappa_{chem}$ analysis was performed using three different schemes to represent hygroscopcity ($\kappa_{chem}$ calculated from ACSM composition + BC, $\kappa_{chem}$ calculated from ACSM composition only and fixed $\kappa_{chem}$=0.3). The median hygroscopicity across sites ranged from approximately 0.2 to 0.5 and increased systematically as the organic fraction decreased.

Aerosol optical properties across the seven sites reveal clear environmental differences. Both $\sigma_{sp}$ and $\sigma_{ap}$ vary with aerosol loading and sources, with continental sites having the highest absorption due to biomass burning and anthropogenic emissions.
At the marine site ENA, high $\sigma_{sp}$ reflects the presence of marine aerosols with high scattering efficiency. BSF and SAE indicate a predominance of fine particles at continental and mountain sites, whereas marine and polar sites are dominated by coarser particles. AAE values remain generally consistent across sites with median values of approximately 1.2, indicating that BC is the primary absorbing component. Most sites are dominated by scattering aerosols (SSA > 0.9), with lower SSA observed at the site with the most urban influence.

The joint dataset of CCN, aerosol chemical composition and optical properties have been used to evaluate the ability of different prediction methods to estimate CCN concentrations, using either chemical composition or aerosol optical properties as inputs. Comparing these prediction methods across site types provides a better understanding of biases and uncertainty in CCN concentration estimates when direct CCN measurements are unavailable. When PNSD measurements are available, assuming a fixed $D_{crit}$ for each SS and counting particles larger than this diameter yields a simple estimate with only a slight
underprediction (MRB = –5%). Similarly, assuming a fixed hygroscopicity ($\kappa_{chem}$ = 0.3) provides a straightforward estimate, but it tends to overpredict CCN concentrations (MRB = 24%). When chemical composition measurements are combined with PNSD, or when only AOPs are available, prediction accuracy is similar, particularly when using $\kappa_{chem}$ values derived from measured species or AOP-based models incorporating multiple variables. Both approaches perform similarly well (8 < |MRB| < 27 %). In stations with limited instrumentation, measuring AOPs — especially $\sigma_{sp}$, BSF, and SAE — allows the application
of S2019 parameterization presented here, which performs robustly (MRB = -8%) across environments and SS, and involves fewer assumptions than chemically-based methods.

The random forest model approach allowed investigation of a wider range of AOPs than included in the S2019 parameterization. Our RF analysis also represents, to the best of our knowledge, the first time the absorption Ångström exponent (AAE) has been explicitly considered as a predictor in CCN estimation based on aerosol optical properties. The random forest model indicated





the importance of AAE in the prediction of the Twomey exponent $k$, highlighting the potential of including absorbing aerosol characteristics in future parametrizations.

Both the empirical (Shen-based) and machine learning (random forest) approaches presented here offer a pathway to estimate long-term trends in CCN concentrations at stations with extensive archives of aerosol optical data. Applying these methods retrospectively could provide insights into the evolution of aerosol-cloud interactions over recent decades. However, a key

requirement for such analyses is a robust quantification of the associated prediction uncertainties, which will be essential to ensure the reliability of inferred trends.

Finally, while this study adds to the accumulated knowledge and previous synthesis of data (e.g., Schmale et al., 2018) relevant for CCN analysis, there are still gaps in spatial coverage. Other observational sites making PNSD and CCN measurements do exist. A truly global CCN climatology, similar in spirit to the effort of Rose et al. (2021) for $N_{tot}$ and PNSD, would require an

extensive harmonization of disparate datasets - it would be a monumental but valuable undertaking.

## Appendix A:  Overview of S2019 methodology

The first step in the approach of S2019 demonstrates that a logarithmic function more accurately captures the dependence of $N_{CCN}$ on SS than other commonly used fits (see Fig. 1 in S2019). Figure S2 in the Supplementary Material shows the same result for the stations considered here. The second step explores the relationship between $N_{CCN}$ and $\sigma_{sp}$, highlighting the

role of BSF in modulating this dependence. S2019 introduce the ratio $R_{CCN/\sigma} = N_{CCN}/\sigma_{sp}$ and show that there is a linear relationship between $R_{CCN/\sigma}$ and BSF:

$$R_{\mathrm{CCN}/\sigma} = \frac{N_{\mathrm{CCN}}}{\sigma_{\mathrm{sp}}} = a \cdot \mathrm{BSF} + b \tag{A1}$$

Equation A1 provides the starting point for the parameterization of CCN using aerosol optical properties. Fit coefficients at each SS for the sites analyzed in S2019 are listed in their Table 3, while those for the sites in this study are shown in Table S1.

This relationship clearly differs among sites and for different SS. To eliminate the SS dependence, the slopes ($a_{\mathrm{SS}}$) and offsets ($b_{\mathrm{SS}}$) from the linear regressions are plotted against the SS, following the S2019 methodology. As shown in Figure A1, the data follow a logarithmic fit, leading to the reformulation of equation A1 as:

$$N_{\mathrm{CCN}} = (a_{\mathrm{SS}} \cdot \mathrm{BSF} + b_{\mathrm{SS}}) \cdot \sigma_{\mathrm{sp}} = ((a_1 \ln(\mathrm{SS}) + a_0) \cdot \mathrm{BSF} + b_1 \ln(\mathrm{SS}) + b_0) \cdot \sigma_{\mathrm{sp}} \tag{A2}$$

The coefficients $a_1$, $a_0$, $b_1$ and $b_0$ with their respective errors from both this study and Shen et al. (2019) are shown in Table

A1. Next, to obtain a site-independent parametrization, the different coefficients from all sites are combined. Figure A2 shows the relationships of the coefficients $a_0$ vs. $a_1$, $b_0$ vs. $b_1$, $a_1$ vs. $b_1$, and $a_0$ vs. $b_0$. Linear regressions yield $a_0 = (2.41 \pm 0.13)a_1$, $b_0 = (2.42 \pm 0.12)b_1$ and $b_1 = (-0.095 \pm 0.011)a_1 + (5.7 \pm 11.0)$. Considering these relationships and, after the development





shown in Section "Derivation of equation A3 " in the Supplement, equation A2 can be expressed as:

$$N_{\text{CCN}} \approx \ln\left(\frac{\text{SS}}{0.089 \pm 0.011}\right)\left[a_1\left(\text{BSF} - (0.095 \pm 0.011)\right) + (5.7 \pm 11.0)\right]\sigma_{\text{sp}} \tag{A3}$$

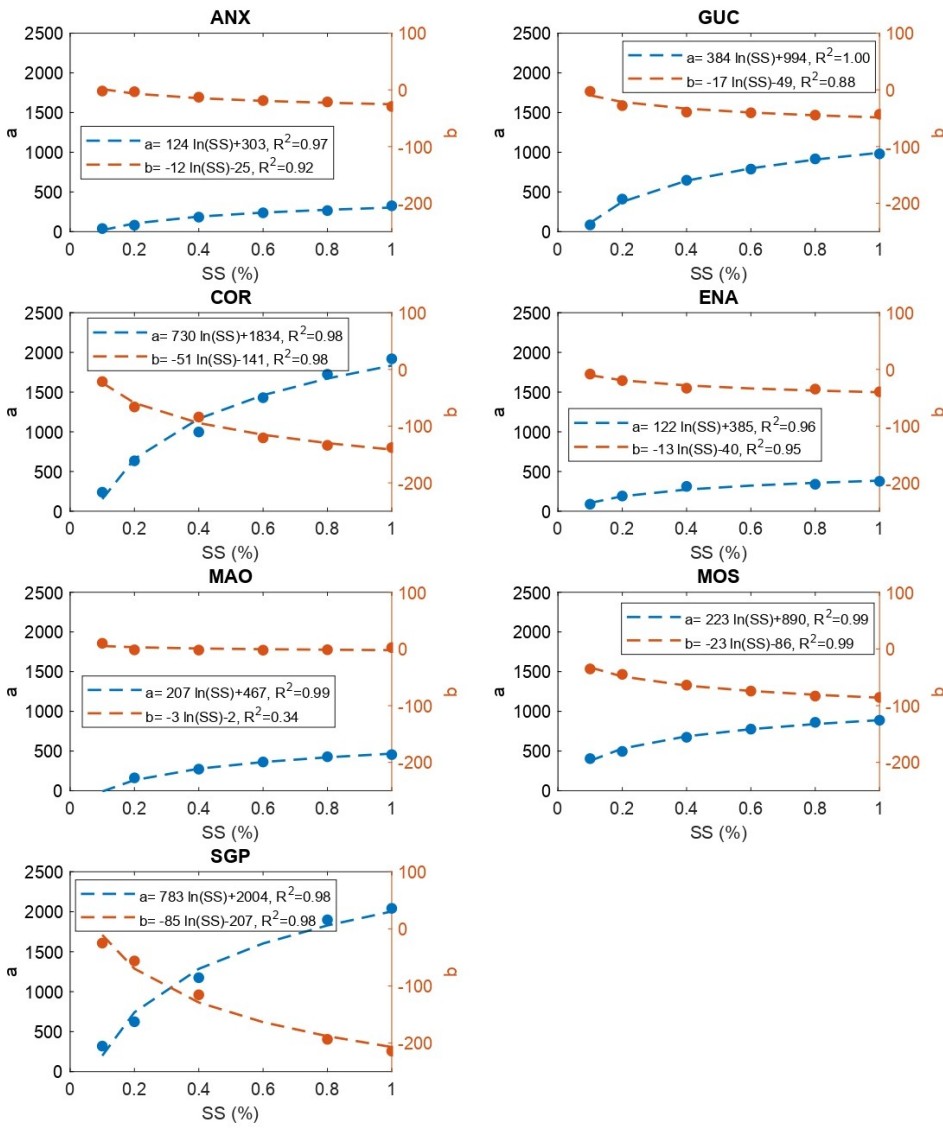

**Figure A1.** Slopes (a) and offsets (b) of the linear regressions $R_{CCN/\sigma} = a.BSF + b$ of each site (Table S1) as a function of SS. Logarithmic fitting applied to data.



**Table A1.** Coefficients $a_1$, $a_0$, $b_1$ and $b_0$ from the logarithmic fitting of coefficients in Table S1 to SS. Shen et al. values are given in Table 4 of Shen et al. (2019). SE: standard error of the respective coefficient obtained from the linear regressions.

|  | **Site** | $a_1 \pm$ SE | $a_0 \pm$ SE | $b_1 \pm$ SE | $b_0 \pm$ SE |
|---|---|---|---|---|---|
| Shen et al. | SMEAR II | $464 \pm 11$ | $1170 \pm 16$ | $-49 \pm 1.5$ | $-118 \pm 0.67$ |
| | SORPES | $331 \pm 12$ | $817 \pm 18$ | $-26 \pm 0.9$ | $-62 \pm 1.4$ |
| | PGH | $205 \pm 30$ | $385 \pm 41$ | $-6.3 \pm 1.5$ | $-9.1 \pm 2.0$ |
| | PVC | $810 \pm 17$ | $1933 \pm 21$ | $-70 \pm 1.7$ | $-160 \pm 2.1$ |
| | MAO | $393 \pm 45$ | $858 \pm 40$ | $-25 \pm 6.6$ | $-60 \pm 5.8$ |
| | ASI | $52 \pm 17$ | $164 \pm 26$ | $-2.9 \pm 1.6$ | $-6.3 \pm 2.3$ |
| This work | ANX | $124 \pm 18$ | $303 \pm 14$ | $-11 \pm 2.9$ | $-25 \pm 2$ |
| | GUC | $384 \pm 20$ | $994 \pm 17$ | $-17 \pm 5$ | $-49 \pm 4$ |
| | COR | $730 \pm 96$ | $1834 \pm 77$ | $-51 \pm 6$ | $-141 \pm 5$ |
| | ENA | $122 \pm 30$ | $385 \pm 23$ | $-13 \pm 4$ | $-40 \pm 3$ |
| | MAO | $207 \pm 16$ | $467 \pm 13$ | $-3 \pm 4$ | $-2 \pm 3$ |
| | MOS | $222 \pm 23$ | $889 \pm 18$ | $-23 \pm 2$ | $-86 \pm 2$ |
| | SGP | $783 \pm 140$ | $2003 \pm 106$ | $-85 \pm 16$ | $-206 \pm 12$ |

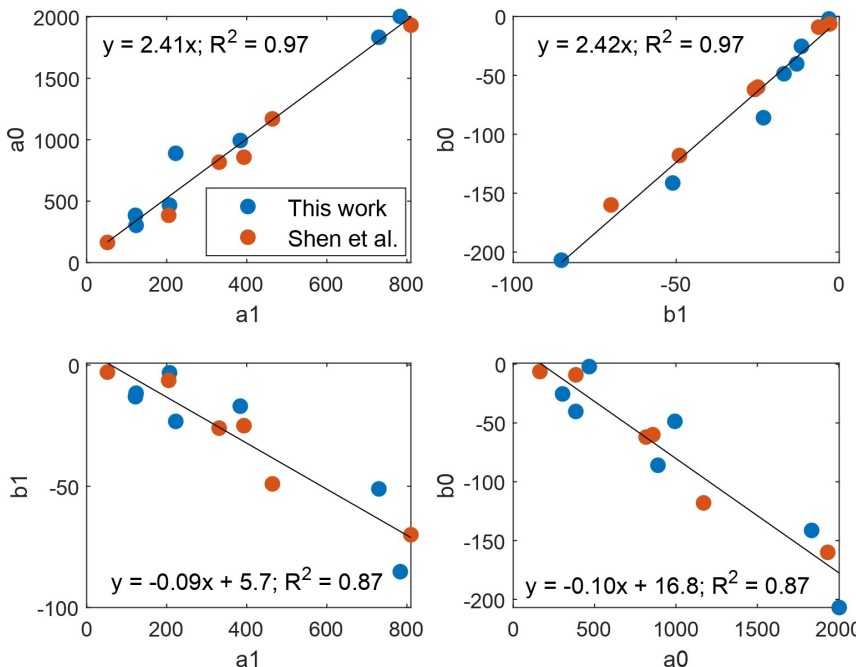

**Figure A2.** Relationship between the coefficients $a_0$, $a_1$, $b_0$, and $b_1$ of Eq. A2 for each site shown in Table A1. The coefficients units are $cm^{-3}$ Mm.

It was shown in Shen et al. (2019) that when the number of hourly samples exceeds approximately 1000 — a condition also met at all our sites — the uncertainty in the minimum BSF (BSF$_{min}$) becomes sufficiently low. Therefore, instead of subtracting



a fixed offset of $(0.095 \pm 0.025)$ from the BSF, we use the observed minimum BSF value (BSF$_{\min}$; 1st percentile of BSF). In addition, as shown in the derivation presented in Supplementary Section S4 of Shen et al. (2019), the final term $(5.7 \pm 11.0)$ is treated as a constant C, which depends on $R_{min}$, defined as the minimum (first percentile) of $N_{CCN}/\sigma_{sp}$. Taking all this into account, Eq. A3 can be reformulated by incorporating these terms, and is written as follows.

$$N_{\text{CCN}} \approx \left( a_1 \ln \left( \frac{\text{SS}}{0.089 \pm 0.011} \right) (\text{BSF} - \text{BSF}_{\min}) + R_{\min} \right) \cdot \sigma_{\text{sp}}. \tag{A4}$$

The final step consists of relating the coefficient $a_1$ in Eq. A4 to the scattering Ångström exponent (SAE), which is the only parameter among optical properties found to be positively correlated with $a_1$. Based on the median values from Shen et al. (2019) and from this study, linear regression yields $a_1 \approx (320 \pm 78) \cdot \text{SAE cm}^3$ Mm (Fig. A3). Additionally, the minimum value of $R$ in Eq. A4, $R_{\min}$, was estimated as the 1st percentile of $R_{CCN/\sigma}$ at each site and supersaturation, resulting in an average value of $R_{\min} = 8.7 \pm 9.3$ cm$^{-3}$ Mm. Consequently, the parameterization becomes

$$N_{\text{CCN}} \approx \left[ (320 \pm 78) \, \text{SAE} \cdot \ln \left( \frac{\text{SS}}{0.089 \pm 0.011} \right) (\text{BSF} - \text{BSF}_{\min}) + (8.7 \pm 9.3) \right] \cdot \sigma_{\text{sp}}. \tag{A5}$$

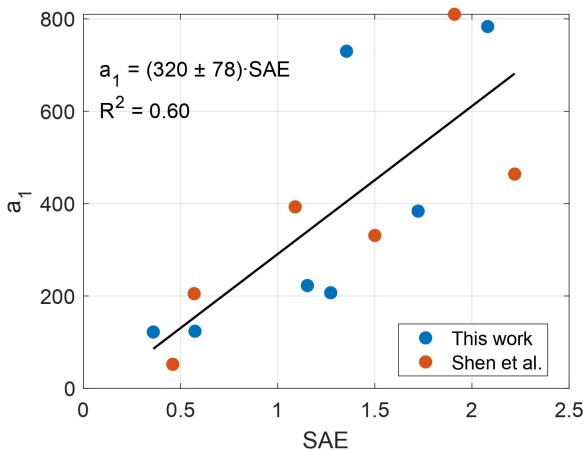

**Figure A3.** Relationship of the $a_1$ coefficient in Eq. A3 with the average PM$_{10}$ scattering Ångström exponent (SAE).

*Code availability.* Code will be made available on request.

*Data availability.* All data presented here are described in Andrews et al. (2025a) and accessible at Andrews et al. (2025b).



*Author contributions.* I.Z. wrote the original draft, performed visualization, investigation, formal analysis, data curation, and conceptualization. J.A.C.V. contributed to writing – review & editing, methodology, investigation, formal analysis, conceptualization, and supervision. E.A. contributed to writing – review & editing, methodology, investigation, and conceptualization. A.C. contributed to data curation, conceptualization, writing – review & editing,. G.C.-C. contributed to writing – review & editing. A.G.H. contributed to writing – review & editing and funding acquisition. G.T. contributed to writing – review & editing, supervision, project administration, methodology, funding 830 acquisition, and conceptualization.

*Competing interests.* A.G.H. serves on the Editorial Board of Atmospheric Chemistry and Physics. The other authors declare no competing interests.

*Acknowledgements.* We thank contribution from MICIU/AEI /10.13039/501100011033/ and "European Union NextGenerationEU/PRTR" via NUCLEUS project PID2021-128757OB-I00, the University of Granada Scientific Unit of Excellence: Earth System (UCE-PP2017-02) 835 and the MIXDUST project (PID2024-160280NB-I00) funded by MICIU/AEI /10.13039/501100011033/ and by FEDER, EU. We acknowledge the DOE/ARM mentors for providing help with data issues.

*Financial support.* This work was supported through DOE/ASR funding via grant number DE-SC0022886.



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
