# Peer review of "Cloud condensation nuclei phenomenology: predictions based on aerosol chemical and optical properties"

_EGUsphere, 2025_

## Referee Comment (RC2)

This study draws on observational data of aerosol chemical composition, particle size distributions, aerosol optical properties, and cloud condensation nuclei (CCN) number concentrations from ten sites across diverse environments. CCN concentrations are predicted using two approaches: (1) a combination of chemical composition and particle size distributions, and (2) optical properties alone. The results based on optical parameters─derived using both traditional empirical formulas and machine-learning methods─are particularly noteworthy and offer valuable insights.

Overall, the manuscript has the potential to be published in Atmospheric Chemistry and Physics (ACP). However, despite the relatively detailed Methods section, several essential methodological and contextual details remain unclear, and the organization of the manuscript would benefit from further refinement. I recommend publication after the authors thoroughly address the following major comments and substantially revise the manuscript.

**Major comments:**

1. I believe the primary emphasis of the manuscript should be the section where CCN number concentrations are predicted from aerosol optical properties. The earlier analysis based on chemical composition and particle size distributions using the κ-Köhler theory could be substantially streamlined. The configurations of Scheme 1 and Scheme 2 provide limited additional insight, as the inclusion of black carbon will inevitably reduce overall aerosol hygroscopicity. The authors note that excluding black carbon is common in previous studies, but this likely reflects the absence of black carbon measurements in those datasets; in contrast, many studies that do measure black carbon appropriately include it in their calculations. Scheme 3 is more relevant, as it helps assess potential biases that arise when models apply a uniform hygroscopicity parameter across different environments. Therefore, I recommend focusing only on Scheme 1 and Scheme 3 and providing a shorter, more concise, discussion of this part of the analysis.

2. In the Random Forest prediction section, the authors should report model performance for both the training and test datasets—such as the coefficient of determination—to assess potential overfitting or underfitting. Demonstrating model reliability is necessary before conducting deeper analysis. The authors should also include a table summarizing the key training parameters for each model, such as the number of trees and maximum tree depth.

For model validation, I recommend using k-fold cross-validation rather than relying on a single train–test split. Finally, a flowchart illustrating the different models will help readers more clearly understand the input features and output structure of each approach.

3.  The authors base their conclusions of the model performance only on the MSB. However, they should also include a metric regarding the precision, such as RSME. For example, the new equation has a lower bias than the RF models, but e.g. comparing Figs 7b and Fig 9, it might have a larger spread in the predicted vs measured values.

4.  The authors introduce a new fitting equation for predicting CCN number concentrations from aerosol optical parameters, distinct from the formulation in Shen et al. (2019). This is an important contribution, and the derivation of the new equation should be presented more clearly to improve reader understanding. Currently, the descriptions of both the Shen et al. (2019) equation and the new equation are fragmented, with some content placed in the appendix and additional formulas included below Table S1 in the Supplement. I recommend reorganizing and integrating these materials by presenting the explanation of the Shen et al. (2019) equation alongside the derivation of the new equation in a single, coherent section. This could be either done in the main text or the SI. This restructuring would enhance clarity and help readers follow methodological development more effectively.

5.  I find that there is some redundancy between the Discussion and Conclusion sections. For example, the description of different methods in previous studies (lines 679–689 in the Discussion) should be more appropriately placed in the Introduction. Similarly, the summary of the current study's results after line 702 overlaps with content in the Conclusion. I suggest that the authors reorganize and consolidate these sections to make the manuscript more concise and coherent.

6.  Page 14, Figure 2/Table 1. The activation diameters at the ASI site seem very low, with the mode of the frequency distribution at 40 nm. For a supersaturation of 0.4%, this corresponds to pure NaCl particles (pure ammonium sulfate particles would have an activation diameter of around 50 nm for S = 0.4%). Since even lower activation diameters down to 30 nm are routinely observed, this seems unlikely and might point to a potential problem with the data. Please do additional quality control and explicitly discuss this

issue in the manuscript.

7. Throughout the manuscript the activation diameters are referred to as "critical diameters". This usage has unfortunately become more common in the literature. Usually in Köhler theory the term "critical diameter" is used for the ambient particle diameter at activation (i.e. the diameter indicating the maximum of the Köhler curve, corresponding to Scrit). I would encourage the use of another term, such as "(dry) activation diameter", but I leave this up to the authors.

**Minor comments:**

1. Line 29: The classification of site types in this section seems confused. Urban and high-altitude sites are generally considered part of the continental region. The authors should revise this description for greater accuracy.

2. Line 144: The authors frequently refer to figures in the supplement of Andrews et al. (2025a). I suggest that the authors reproduce these validation plots using the original data and include them in the appendix of the current manuscript. Requiring readers to consult the appendix of another paper is inconvenient and may hinder understanding.

3. Line 164: It is unclear how the authors determined the critical diameter—was it measured using combined CCNc and SMPS, or obtained by another method? The authors should clarify this and provide the relevant calculation formulas in the main text or the Supplement.

4. Line 376: Here the authors use "$SO_4^{2-}$, $NO_3^-$…", while they use "sulfate, nitrate…" in the Figure 3.

5. Line 379: The font of the symbols "$\mu g/m^3$" here differs from that used in line 383.

6. Figure 4b: The denominator "IA+OA" is missing parentheses.

7. Figure 5: Please specify in the figure caption how the number of points in the filled areas was determined.

8. Line 496: The citation format here is incorrect; it should be "Saliba et al., 2020." Please check for similar errors elsewhere in the manuscript.

9. Figure 7, Figure 8c, and Figure 9c: The same question for Figure 5.

10. Figure 10: Please show colorbar for this heatmap.